# Rethinking Variational Inference for Probabilistic Programs with Stochastic Support

**Tim Reichelt[1]**    **Luke Ong[1,2]**    **Tom Rainforth[1]**

[1] University of Oxford
[2] Nanyang Technological University, Singapore
{tim.reichelt,lo}@cs.ox.ac.uk    rainforth@stats.ox.ac.uk

## Abstract

We introduce *Support Decomposition Variational Inference* (SDVI), a new variational inference (VI) approach for probabilistic programs with stochastic support. Existing approaches to this problem rely on designing a single global variational guide on a variable-by-variable basis, while maintaining the stochastic control flow of the original program. SDVI instead breaks the program down into sub-programs with static support, before automatically building separate sub-guides for each. This decomposition significantly aids in the construction of suitable variational families, enabling, in turn, substantial improvements in inference performance.

## 1  Introduction

Probabilistic programming systems (PPSs) enable users to express probabilistic models with computer programs and provide tools for inference. Many PPS, such as Stan [1] or PyMC3 [2], limit the expressiveness of their language to ensure that the programs in their language always correspond to models with static support—i.e. the number of variables and their support do not vary between program executions. In contrast, *universal PPSs* [3–11] can encode programs where the sequence of variables itself—not just the variable values—changes between executions, leading to models with stochastic support. These models have applications in numerous fields, such as natural language processing [12], Bayesian Nonparametrics [13], and statistical phylogenetics [14]. A wide range of simulator-based models similarly require such stochastic control flow [15–17].

The effectiveness of PPSs is heavily reliant on the underlying inference schemes they support. Variational inference (VI) is one of the most popular such schemes, both in PPSs and more generally [18–20]. This popularity is due to its ability to use derivatives to scale to large datasets and high-dimensional models [21–24], often providing much faster and more scalable inferences compared to Monte Carlo approaches [25]. To provide the required derivatives, a number of modern universal PPSs—such as Pyro [5], ProbTorch [26], PyProb [15], Gen [7], and Turing [6]—have introduced automatic differentiation [27] capabilities for programs with stochastic control flow. One of the core aims behind these developments was to support VI schemes in such settings [5].

However, constructing appropriate variational families, typically known as guides in PPSs, can be very challenging for problems with stochastic support, even for expert users. This is because the stochasticity of the control flow induces discontinuities and complex dependency structures that are difficult to remain faithful to and design parameterized approximations for. Furthermore, while there are a plethora of different automatic guide construction schemes for static support problems [18–20], there is a lack of suitable schemes applicable to models with stochastic support. Consequently, existing methods tend to give unreliable results in such settings, as we demonstrate in Figure 1.

We argue that a significant factor of this shortfall is that standard practice—for both manual and automated methods—is to construct the guide on a variable-by-variable basis [28–31]. Namely,

36th Conference on Neural Information Processing Systems (NeurIPS 2022).

```
def model():
    x = sample("x", Normal(0, 1))
    if x < 0:
        z = sample("z1", Normal(-3, 1))
    else:
        z = sample("z2", Normal(3, 1))
    sample("y", Normal(z, 2), obs=2.0)
```

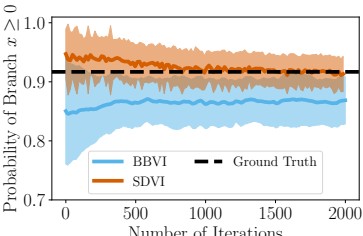

Figure 1: Pyro program with stochastic control flow [Left]. Existing procedures for automatically constructing the guide mirror the control flow of the input program [BBVI, Blue]. However, this produces an inherently limited variational family, leading to unsatisfactory performance despite the problem's simplicity. By breaking down the guide over paths, SDVI [Red] is able to provide accurate inference. Results computed over $10^2$ replications, plotted are mean and standard deviation.

existing approaches generally use a single global guide that mirrors the control flow of the input program, then introduce a variational approximation for each unique variable. This is problematic because control flows inherently introduce discontinuities into the program's density function, such that the conditional distribution of each variable will typically change significantly whenever the program path—that is the sequence of random variables—changes. Thus it is extremely challenging to learn a single approximation for each variable that is appropriate across all paths. Further, as the set of variables that exist can itself be stochastic, it is difficult for such guides to appropriately condition on previously sampled variables. Existing automated approaches, therefore, typically rely on mean-field assumptions [29], thereby forgoing any conditioning on the program path itself, consequently leading to poor approximations for most problems.

To overcome these difficulties, we propose *Support Decomposition Variational Inference* (SDVI), a new VI approach based around a novel way of constructing the variational guide. SDVI "rethinks" the guide construction by breaking it down over *paths*, instead of building it on a variable-by-variable basis. Specifically, by utilizing the fact that any program can be reformulated as a mixture of *straight-line programs* (SLPs) [32–34]—each defined by a unique realization of the path—SDVI constructs the guide as a mixture of sub-guides with static support. We show that optimizing the variational objective with this guide structure leads to a natural decomposition of the overall optimization problem into independent sub-problems, each taking the form of a VI with static support. The sub-guides can thus be effectively constructed and trained using more standard VI techniques, before being recombined to form our overall variational approximation. To make SDVI accessible to a wide audience, we have implemented it in Pyro [5]. We evaluate it on a set of example problems with synthetic and real-world data, finding that it provides substantial improvements over existing techniques.

## 2 Background

### 2.1 Probabilistic Programs in Universal PPSs

PPSs allow users to express probabilistic models and condition on observed data [31, 35]. A common mechanism to achieve this is to extend standard programming languages with two new primitives: `sample` and `observe`.[1] `sample(id, dist)` draws samples from the distribution object `dist`, where `id` is a unique lexical identifier. `observe(id, data, dist)` enables conditioning on an observed outcome `data`, where `dist` and `id` are as before. For problems admitting a Bayesian formulation, the `sample` and `observe` terms can informally be thought of as prior and likelihood factors respectively.

*Universal* PPSs allow users to write complex models whose support can vary from one execution to the next, e.g. stochastic branching can mean certain variables only sometimes exist. This can substantially complicate the process of performing inference.

A probabilistic program in a universal PPS defines an unnormalized *density function* $\gamma(x_{1:n_x})$ over the *raw random draws* $x_{1:n_x} \in \mathcal{X}$—defined as the (sequences of) direct outputs of `sample` statements—where $n_x \in \mathbb{N}^+$ is itself potentially random. Though each outcome of $x_{1:n_x}$ uniquely defines a program execution, it is notationally convenient to further associate an *address* $a_i$ to each draw $x_i$ that indicates the position in the program the draw was made. This address can be uniquely defined as the tuple formed by the `id` of the `sample` and the number of times that `sample` has previously been called. For a given execution of the program, the addresses now form an *address path* $A = a_{1:n_x}$.

---

[1]Pyro instead overloads the `sample` primitive: `sample(id,dist,obs=data)` ≡ `observe(id,data,dist)`.

Each `sample` statement encountered during execution contributes the factor $f_{a_i}(x_i \mid \eta_i)$ to the program density, where $a_i$ is the address of the `sample` statement, $f_{a_i}$ is a parameterized density function, and $\eta_i$ are its associated parameters. Similarly, each encountered `observe` statement contributes the factor $g_{b_j}(y_j \mid \phi_j)$, with $b_j$ denoting an address, $y_j$ the observed value, $g_{b_j}$ a parameterized density function, and $\phi_j$ its parameters. Following [36, §4.3.2], we write the program density function as

$$\gamma(x_{1:n_x}) := \prod_{i=1}^{n_x} f_{a_i}(x_i \mid \eta_i) \prod_{j=1}^{n_y} g_{b_j}(y_j \mid \phi_j). \tag{1}$$

All of $n_x, n_y, a_{1:n_x}, \eta_{1:n_x}, y_{1:n_y}, b_{1:n_y}$, and $\phi_{1:n_y}$ are potentially random variables. The goal of *inference* is to approximate the conditional distribution of the program, which has normalized density $\pi(x_{1:n}) = \gamma(x_{1:n})/Z$ with marginal likelihood $Z = \int_{\mathcal{X}} \gamma(x_{1:n_x})dx_{1:n_x}$ and the integral is computed with respect to a reference measure that is implicitly defined by the `sample` statements in the program.

## 2.2 Variational Inference

Variational Inference (VI) [18, 37] solves the inference problem by transforming it to an optimization problem. Specifically, given an unnormalized joint distribution $\gamma(x)$ and a parameterized distribution $q(x; \phi)$, VI computes the variational parameters $\phi$ such that $q(x; \phi)$ most closely approximates $\pi(x) = \gamma(x)/Z$. This is most commonly done by maximizing the *Evidence Lower Bound* (ELBO) $\mathcal{L}(\phi) := \mathbb{E}_{q(x;\phi)}\left[\log \gamma(x)/q(x;\phi)\right]$ via stochastic gradient ascent using Monte Carlo estimates of $\nabla_\phi \mathcal{L}(\phi)$ [38]. Two popular estimators are the score function estimator [39, 40], and the reparameterized gradient estimator [24, 41, 42]. The latter provides lower variance gradient estimates but requires that the distribution $q(x; \phi)$ can be reparameterized and that $\gamma(x)$ is differentiable everywhere.

## 3 Difficulties for Variational Inference in Universal PPSs

The starting point for any VI scheme is to construct an appropriate variational family, also known as a guide. To automate inference, we desire to (at least partially) automate the process of constructing this guide. Existing methods for this all generate the guide on a variable-by-variable basis [28, 29, 31]: they introduce a single variational distribution $q_{a_i}(x; \phi_{a_i})$ for each unique sampling address $a_i$, then form the guide by replacing all the original random draws, $x_i \sim f_{a_i}(\cdot \mid \eta_i)$, with draws from the corresponding variational distribution instead. This forms a global guide that maintains the stochastic dependency structure of the original program, such that the guide itself has stochastic support.

Our motivating insight is that this high-level approach has some fundamental limitations. Consider the simple example from Fig. 1. Here the variable x influences the program's control flow. This causes discontinuities that mean it is difficult to approximate its conditional density with a single variational approximation, especially if that variational approximation is restricted to a simple distribution class. Here the different possible paths are essentially working against each other, as what is helpful for the approximation of x on one path, is generally detrimental for the other.

A further complication occurs when the stochastic control flow of a program influences whether a variable exists at all. Here it can become extremely challenging to set up guides which are faithful to the dependency structure of previously sampled variables [43], as the set of variables we condition on is itself stochastic. Because of this, existing approaches typically rely on mean-field assumptions across paths [28–30]. However, this assumption is rarely reasonable given that the path typically strongly influences the distribution of individual variables.

Finally, creating a single unique variational approximation for each address also leads to challenging optimization problems: the same address can be present in multiple program paths, and the number of variables and their dependencies can vary between different paths.

## 4 Support Decomposition Variational Inference

We now introduce a novel VI approach to overcome the challenges mentioned in Sec. 3. We call our method *Support Decomposition Variational Inference* (SDVI), because the key design decision is the choice (and automatic construction) of a guide that takes the form of a mixture distribution over the set of possible paths a program can take. That is, rather than constructing a single guide with the same stochastic control flow as the original program and a separate variational approximation for

each *unique address*, we instead construct separate sub-guides with deterministic control flows for each *unique path*. These are then combined into an overall guide using the mixture distribution which maximizes the overall ELBO. As we will see, this alternative approach substantially simplifies the process of constructing effective guides and allows the full weight of the well-developed techniques for VI in the static support setting to be brought to bear on problems with stochastic support.

## 4.1 Decomposing Probabilistic Programs into Straight-Line Programs

As noted by, e.g., [32–34], all probabilistic programs can be reformulated as mixture distributions over *straight-line programs* (SLPs), which are sub-programs without any stochastic control flow. Building on our earlier notation, the constituent SLPs of a program correspond directly to the unique instances of the program path, $A$. Given the path, the set of variables making up the raw random draws in the program is fixed, along with their form and reference distribution; that is each SLP represents a probabilistic model with fixed variable typing and support.

Following the notation of Zhou et al. [34], we can apply an arbitrary fixed ordering on the set of SLPs in a program, such that we can uniquely define and index them using the set of possible addresses $A_k$ for $k \in \mathcal{K}$, where $\mathcal{K}$ is a countable (but potentially infinite) indexing set. Each SLP $A_k$ now corresponds to a particular sub-region, $\mathcal{X}_k$, of the raw random draw sample space, $\mathcal{X}$. These sub-regions are disjoint and their union is the full sample space. Unlike $\mathcal{X}$, each element in any given $\mathcal{X}_k$ has the same length $n_k$ and is measurable with respect to the same reference measure. The unnormalized density for the $k$th SLP is now given by

$$\gamma_k(x_{1:n_k}) = \mathbb{I}[x_{1:n_k} \in \mathcal{X}_k] \, \gamma(x_{1:n_k}) = \mathbb{I}[x_{1:n_k} \in \mathcal{X}_k] \prod_{i=1}^{n_k} f_{A_k[i]}(x_i \mid \eta_i) \prod_{j=1}^{n_y} g_{b_j}(y_j \mid \phi_j), \quad (2)$$

and the unnormalized density function for the original program can be written as a simple sum of the individual SLP densities: $\gamma(x_{1:n_x}) = \sum_{k \in \mathcal{K}} \gamma_k(x_{1:n_x})$. The corresponding normalized conditional density can then be written as mixture distribution over the conditional distributions of the individual SLPs, with mixture weights given by their (normalized) local partition functions:

$$\pi(x) = \sum_{k \in \mathcal{K}} \pi(x \mid k) \, \pi(k) \quad \text{where} \quad \pi(x \mid k) = \frac{\gamma_k(x)}{Z_k}, \quad \pi(k) = \frac{Z_k}{\sum_{\ell \in \mathcal{K}} Z_\ell}, \quad Z_k = \int_{\mathcal{X}_k} \gamma_k(x) dx. \quad (3)$$

## 4.2 Decomposing the Variational Family into Straight-Line Programs

The key idea behind SDVI is now to construct the guide using a factorization that is analogous to that of the SLP decomposition above. Precisely, we aim to learn variational approximations of the form

$$q(x; \phi, \lambda) = \sum_{k=1}^{K} q_k(x; \phi_k) q(k; \lambda) \quad (4)$$

where $q(k; \lambda)$ defines a categorical distribution over the indices of the SLPs, with support $k \in \{1, \ldots, K\}$; and $q_k(x; \phi_k)$ is the local guide of the $k$th SLP, with support $x \in \mathcal{X}_k$. Critically, as each $\mathcal{X}_k$ represents a fixed support, the local variational families $q_k$ can be automatically constructed using standard techniques for static problems, as we discuss in Sec. 4.5. Note that it is valid for the guide $q(x; \phi, \lambda)$ to not cover all SLPs, i.e. it is possible that $K < |\mathcal{K}|$.

Writing $\phi = \{\phi_k\}_{k=1}^{K}$, the KL divergence we wish to minimize for standard VI is now

$$\text{KL}(q(x; \phi, \lambda) \parallel \pi(x)) = \mathbb{E}_{q(x; \phi, \lambda)} [\log q(x; \phi, \lambda) - \log \pi(x)], \quad (5)$$

which we call the *global KL divergence*. By standard reasoning, minimizing this is equivalent to maximizing the *global ELBO*

$$\mathcal{L}(\phi, \lambda) = \mathbb{E}_{q(x; \phi, \lambda)} [\log \gamma(x) - \log q(x; \phi, \lambda)] \quad (6)$$

which, as we show in App. A, can be rewritten as

$$\mathcal{L}(\phi, \lambda) = \mathbb{E}_{q(k; \lambda)} [\mathcal{L}_k(\phi_k) - \log q(k; \lambda)], \quad \text{where} \quad \mathcal{L}_k(\phi_k) := \mathbb{E}_{q_k(x; \phi_k)} \left[ \log \frac{\gamma_k(x)}{q_k(x; \phi_k)} \right] \quad (7)$$

is the term we refer to as the *local ELBO* for the $k$th SLP. Notice that each $\mathcal{L}_k(\phi_k)$ depends only on the parameter $\phi_k$ and the local SLP density $\gamma_k$; it is completely independent of $q(k; \lambda)$, the other SLPs, and $\phi_{k'}$ for $k' \neq k$. Thus, it follows from (7) that the inference problem for the whole program can be decomposed into independent 'local' inference problems for the component SLPs, along with establishing the mixture probabilities $q(k; \lambda)$. Furthermore, it turns out that the optimal $q(k; \lambda)$ is simply the softmax of $\mathcal{L}_1, \ldots, \mathcal{L}_K$, as shown by the following result.

---

**Algorithm 1** Support Decomposition Variational Inference

---

**Require:** Target program $\gamma$, iteration budget $T$, minimum no. of SH candidates $m$
1: Extract SLPs $\{\gamma_k\}_{k=1}^K$ from $\gamma$ and set $\mathcal{C} = \{1, \ldots, K\}$         $\triangleright$ Sec 4.3
2: Formulate guide $q_k$ for each SLP and initialize parameters $\phi_k$       $\triangleright$ Sec 4.5
3: **for** $l = 1, \ldots, L = \lceil \log_2(K) - \log_2(m) + 1 \rceil$ **do**
4:      **for** $k \in \mathcal{C}$ **do**
5:          Perform $\lfloor T/L|\mathcal{C}| \rfloor$ optimization iterations of $\phi_k$ targeting $\mathcal{L}_{\mathrm{surr},k}(\phi_k)$    $\triangleright$ Sec 4.5
6:      **end for**
7:      Remove $\min(\lfloor |\mathcal{C}|/2 \rfloor, |\mathcal{C}| - m)$ SLPs from $\mathcal{C}$ with the lowest $\mathcal{L}_k(\phi_k)$    $\triangleright$ Sec. 4.4
8: **end for**
9: Truncate $q_k$ outside of SLP support, $\mathcal{X}_k$, using Eq. (13)
10: Estimate each $\mathcal{L}_k(\phi_k)$ using Monte Carlo estimate of Eq. (7)
11: Calculate $q(k; \lambda)$ according to Eq. (8) and return $q(x; \phi, \lambda)$ as per Eq. (4)

---

**Proposition 1.** *Let $L = \{\mathcal{L}_1, \ldots, \mathcal{L}_K\}$ be the set of local ELBOs, defined as per (7), where $L$ is countable but potentially not finite. If $0 < \sum_{k=1}^K \exp(\mathcal{L}_k) < \infty$, then the optimal corresponding $q(k; \lambda)$ in terms of the global ELBO (6) is given by*

$$q(k; \lambda) = \exp(\mathcal{L}_k) \Big/ \sum_{\ell=1}^K \exp(\mathcal{L}_\ell). \tag{8}$$

The proof of this result is given in App. A. Though each of the $\mathcal{L}_k$ terms here is itself intractable, they can be estimated efficiently and accurately by simple Monte Carlo. We can thus straightforwardly construct $q(k; \lambda)$ once we have learned our local variational approximations: noting that these two processes are separable, $q(k; \lambda)$ is not needed until *after* the individual $q_k$ are trained.

### 4.3 Finding SLPs

We have just shown how we can solve the VI problem of a probabilistic program with stochastic support by reducing it to a set of *independent* and *simpler* VI problems, each concerning an SLP, a program with static support. However, we still need a mechanism to 'discover' the SLPs, i.e. extract the possible address paths from a program.

Here we first note that we only need to consider an SLP if it has a non-zero probability of being identified under forward simulation of the program, while ignoring conditioning statements. This hints at a cheap and simple discovery mechanism whereby we draw samples by forward simulation and take note of the unique paths that have been generated. We can either do this upfront, or in an online manner whereby we seek new SLPs as our budget increases and we have scope to deal with them (see App. B for details). Although this is a stochastic process that is not guaranteed to find all the SLPs for finite budgets, for the problems considered in our experiments, it was always able to reliably identify all SLPs with *non-negligible posterior mass*. Nonetheless, this approach may not be sufficient for all problems, such as when the likelihood concentrates in an area of very low prior mass. Here one should instead look to employ more sophisticated discovery methods instead, such as those based on MCMC sampling [34] or static analysis of the program code [32, 44, 45].

### 4.4 Allocating Resources

Using the same amount of computational budget on each SLP is potentially wasteful, particularly if there is a large number of SLPs with insignificant marginal likelihoods. Therefore, we seek a scheme that allocates more computational resources to promising SLPs, making sure to exploit the fact that the different inference problems are trivially parallelizable.

To formalize this resource allocation problem, let $T$ represent some fixed resource budget. Further, let $t_k$ be the amount of this budget we spend on optimizing the $k$th SLP, such that $\sum_k t_k = T$ at the end of our training. Our ultimate aim is produce the maximum possible final global ELBO, which will be a function of $\phi_1(t_1), \ldots, \phi_K(t_K)$, where $\phi_k(t_k)$ denotes the value of $\phi_k$ achieved after allocating $t_k$ resources to that SLP. By plugging the optimal mixture distribution $q(k; \lambda)$ from (8) into (6), we see that, after some rearranging, our resource allocation can be formulated as trying to maximize

$$\mathcal{L}(\phi, \lambda^*) = \log \sum_{k=1}^K \exp(\mathcal{L}_k(\phi_k(t_k))) \quad \text{s.t.} \ \sum_{k=1}^K t_k = T. \tag{9}$$

In practice, this is not a suitable objective for controlling our resource allocation directly, as it is still itself a random variable given $t_1, \ldots, t_K$, because the optimization procedure is stochastic. Moreover, we cannot consider its expectation, since the distribution of the $\phi_k(t_k)$ is unknown. However, it does provide insight into how we ideally would like to allocate resources: we want to allocate more resources to SLPs whose exponentiated ELBOs are significant. In particular, we can think of the 'reward' for allocating $\epsilon$ more resources to SLP $k$ as $\exp(\mathcal{L}_k(\phi_k(t_k + \epsilon))) - \exp(\mathcal{L}_k(\phi_k(t_k)))$.

One could now, in principle, formulate the problem as a sequential decision making problem [46]. However, the diminishing nature of the rewards and the fact that they are highly unlikely to be sub-Gaussian, along with the need to allow choosing multiple arms at once for parallelization, mean that setting up such an approach which is effective in practice is likely to be quite challenging.

Instead, we propose a simple heuristic, based on the *Successive Halving* algorithm (SH) [47] (see App. B for a description), an approach commonly used for resource allocation in hyperparameter optimization (HO) [48, 49]. In standard SH, the final objective is to identify and train the single best candidate, whereas ours is to maximize the *sum* of all the local ELBOs. Despite this difference, the use of SH can still be justified by the fact that the distribution over $\exp(\mathcal{L}_k(\phi_k(\infty)))$ will typically be heavily concentrated to a small number of SLPs, often only a single one. Nonetheless, we make a small adaptation to the approach to stop over-focusing on a single SLP: we stop the halving process when a chosen number, $1 \leqslant m \leqslant K$, of the candidates are left, with $m = 1$ corresponding to standard SH. The minimum proportion of the budget allocated to any given candidate by this scheme is $1/(K\lceil \log_2 K - \log_2 m + 1 \rceil)$, so we can use $m$ as a hyperparameter to control how evenly resources are allocated, with $m = K$ corresponding to uniform allocation. This approach is also helpful for parallelization, as we can set $m$ equal to the number of available cores.

Putting everything together, a summary of our SDVI algorithm is given in Algo. 1. In App. B, we further show how this can be extended to an online variant of the approach, wherein we repeatedly run SH using the objective $\exp(\alpha \mathcal{L}_k(\phi_k(t_k)))/t_k$, where $0 < \alpha \leqslant 1$ is a hyperparameter, with smaller values of $\alpha$ encouraging more exploration.

## 4.5 Formulating and Training the Local Guides

In Algo. 1 we assume a mechanism to construct the *local guide* $q_k(x; \phi_k)$ for each SLP specified by path $A_k$. In many situations—notably when the program path is uniquely determined by the sampled values from discrete distributions—it is possible to construct guides $q_k$ that are guaranteed to place support within the sub-region $\mathcal{X}_k$, which, in turn, allows us to use the reparameterized gradient estimator for the gradients of $\mathcal{L}_k(\phi_k)$. Many models encountered in practice, e.g. mixture models [13], have this property. In this case it is possible to eliminate all the variables which influence the control flow by conditioning, effectively setting them to constants; see App. C for further details.

In situations where we cannot easily construct a $q_k$ which places support only within $\mathcal{X}_k$, we need to take care when training our guide. Recall that for path $A_k$ the number of variables $n_k$ and their type sequence is fixed, which allows us to construct an initial guide $\tilde{q}_k$ with correct dimensionality and variable typing. Let the support of this guide be denoted by $\mathcal{X}'_k = \mathrm{supp}(\tilde{q}_k)$. In general, we will have $\mathcal{X}_k \subset \mathcal{X}'_k$, because the control flow in the program imposes additional constraints on each individual variable. Having constructed a guide with $\mathrm{supp}(\tilde{q}_k) = \mathcal{X}'_k$, one might be tempted to optimize $\mathrm{KL}(\tilde{q}_k(x; \phi_k) \parallel \pi(x \mid k))$, but we cannot guarantee the absolute continuity condition (i.e. $\tilde{q}_k(x; \phi_k) = 0$ if $\pi(x \mid k) = 0$), and so, the KL divergence may not be well-defined, giving an ELBO of $-\infty$. To alleviate this issue we temporarily create a new surrogate target density defined as

$$\tilde{\gamma}_k(x_{1:n_k}) := \gamma_k(x_{1:n_k}) + c\,\mathbb{I}[x_{1:n_k} \notin \mathcal{X}_k], \tag{10}$$

for a small positive, finite constant $c$. This surrogate density is used solely for optimizing $\tilde{q}_k(x; \phi_k)$. We train $\phi_k$ to optimize the corresponding surrogate ELBO

$$\mathcal{L}_{\mathrm{surr},k}(\phi_k) := \mathbb{E}_{\tilde{q}_k(x; \phi_k)}\left[\log \tilde{\gamma}_k(x) - \log \tilde{q}_k(x; \phi_k)\right]. \tag{11}$$

We need to be careful to choose an appropriate $c$ that is sufficiently small compared to the values of $\gamma_k(x)$ for $x \in \mathcal{X}_k$, which we ensure by setting $c$ adaptively. During the SLP discovery phase (Line 1 in Algo. 1), we keep track of the smallest density value encountered so far, and call that $d_{min}$. We then set $c = 0.01 d_{min}$ to ensure that the density values for $\tilde{\gamma}_k(x)$ outside of $\mathcal{X}_k$ are significantly below the values of $\tilde{\gamma}_k(x)$ for $x \in \mathcal{X}_k$. Hence, optimizing (11) faithfully optimizes $q_k$ to be a good approximation to $\gamma_k(x)$ while avoiding the issues of infinite ELBO values. While $\tilde{\gamma}_k$ is not a proper

unnormalized density (it will in general not integrate to a finite value) this is not an issue in practice due to the mode-seeking behaviour of optimizing the ELBO.

Unfortunately, the bounds on the support of the SLP inevitably create a discontinuity in the objective. Thus, for fully unbiased gradients we need to use the score function estimator or some extension thereof. However, in some cases, the bias of the reparameterization gradient estimator may be sufficiently small to warrant its use. Note that $\mathcal{L}_{\mathrm{surr},k}(\phi_k)$ retains the desirable property of the standard ELBO that, if the observations are conditionally independent given the latent variables, we can get unbiased estimates of the ELBO using minibatches of the full dataset [19, 22, 42].

Further we need to be careful when initializing $\tilde{q}_k$ as we require it to place sufficient probability mass within $\mathcal{X}_k$ to provide a suitable training signal. To ensure this, we initialize $\phi_k$ by minimizing the *forward* KL divergence between the prior density of the $k$th SLP and $\tilde{q}_k(x; \phi_k)$

$$\mathrm{KL}(\pi_{prior,k}(x) \parallel \tilde{q}_k(x; \phi_k)) \propto \mathbb{E}_{\pi_{prior}(x)}\left[-\mathbb{I}[x \in \mathcal{X}_k]\log \tilde{q}_k(x; \phi_k)\right] \tag{12}$$

where $\pi_{prior}(x_{1:n_x}) := \prod_{i=1}^{n_x} f_{a_i}(x_i \mid \eta_i)$. This objective can be optimized via stochastic gradient descent (cf. App. C). Note that, for the purpose of initialization, we are targeting the prior, and thus we do not have to resort to expensive schemes to estimate the gradients which are necessary if one aims to minimize the forward KL targeting the posterior [50].

So far we have outlined how to train $\tilde{q}_k$ but to evaluate the local ELBOs, $\mathcal{L}_k$, we need to construct a distribution $q_k$ which satisfies the hard constraint $\mathrm{supp}(q_k) = \mathcal{X}_k$. Our solution for this is *truncating* $\tilde{q}_k$ by checking whether specific raw random draws $x'_{1:n_k}$ are valid for the path $A_k$, i.e. whether $\mathbb{I}\left[x'_{1:n_k} \in \mathcal{X}_k\right]$. We can do this by simply executing the program with fixed draws set to $x'_{1:n_k}$ and then noting that the program terminates and follows the address path $A_k$ if, and only if, $x'_{1:n_k} \in \mathcal{X}_k$. Thus, we truncate $\tilde{q}_k$ using

$$q_k(x; \phi_k) = \frac{\tilde{q}_k(x; \phi_k)\mathbb{I}\left[x \in \mathcal{X}_k\right]}{\tilde{Z}_k(\phi_k)}, \quad \text{where} \quad \tilde{Z}_k(\phi_k) = \int_{\mathcal{X}'_k} \tilde{q}_k(x; \phi_k)\mathbb{I}\left[x \in \mathcal{X}_k\right] dx. \tag{13}$$

Hence, $q_k$ is implicitly defined as the output of a rejection sampler with $\tilde{q}_k$ as a proposal. Note, that as we use the surrogate ELBO in (11) when training $\phi_k$, we never need to take gradients through $q_k$ or $\tilde{Z}_k(\phi_k)$, thereby avoiding the significant practical issues this would cause (see App. G). Thus, the local guide $q_k$ (Eq. (13)) is only used for estimating the local ELBOs (Eq. (7)). This is done by first drawing $N$ samples $\{x^{(i)}\}_{i=1}^N$ from $\tilde{q}_k$, then rejecting samples which do not fall into the SLP and estimate $\tilde{Z}_k$ as the acceptance rate of this sampler (i.e. $N_A/N$ where $N_A$ is the number of samples accepted). Using $A$ to denote the set of indices of accepted samples, we form our ELBO estimate as

$$\hat{\mathcal{L}}_k := \frac{1}{N_A}\sum_{i \in A}\log(N_A\,\gamma_k(x^{(i)})) - \log(N\,\tilde{q}_k(x^{(i)}; \phi_k)). \tag{14}$$

Note here that $A$ and $N_A$ are random variables that both implicitly depend on $\phi_k$, which is why we can use this for estimation, but not training.

## 5  Related Work

The vast majority of prior work on deriving automated VI algorithms focuses on the setting of static support [20, 23, 51–55]. Of particular note, [43, 56, 57] also consider using variational families that do not match the dependency structure of the original problem, but they still require static support. More generally, there have been models with stochastic support for which bespoke guides where developed which do not follow the control-flow structure of the input program [58]. However, these custom guides do not leverage the breakdown of the input program into SLPs.

The Divide-Conquer-Combine (DCC) algorithm [34] also exploits the breakdown of the program density into individual SLPs. However, [34] mainly focused on local inference algorithms that are sampling based, especially MCMC. As we showed in Sec. 4 unique challenges and opportunities arise when we consider the breakdown from a variational perspective. Further, our work shows that using a variational family based on SLPs naturally leads to divide-and-conquer style algorithm, due to the resulting separability of the ELBO. One of the most practical differences is that SDVI only requires (exponentiated) ELBOs to be estimated for each SLP, rather than marginal likelihoods. The former can typically be estimated substantially more accurately for a given budget, allowing SDVI to

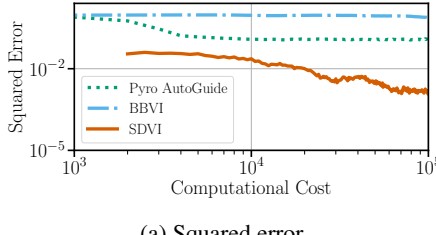
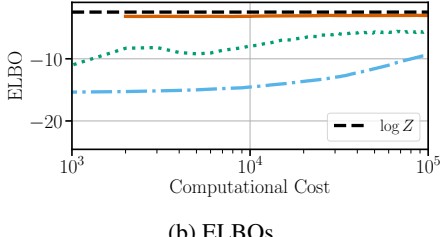

(a) Squared error.                    (b) ELBOs.

Figure 2: Results for the model in § 6.1. Computational cost is measured in the number of likelihood evaluations. For each metric we show the mean and standard deviation over 10 runs. a) Squared error between the true SLP weights and the estimated SLP weights. b) Evidence Lower Bounds (ELBOs) for the variational algorithms, dashed line indicates the analytic log marginal likelihood.

scale better to high dimensional problems (see Sec. 6.2). [32] and [33] both also use the general idea of breaking down programs into SLPs, but both papers consider starkly different problem settings. Neither have any direct link to variational inference.

Our work is situated in the larger context of automated inference for universal PPSs. Other popular approaches include particle-based methods [6, 11, 59–61] and MCMC approaches with automated proposals [1, 62–64]. Some work has looked to perform *amortized* inference over a range of possible datasets [16, 56, 57, 65], often by training a proposal that is similar to a variational approximation.

# 6   Experiments

To make SDVI easily accessible to practitioners we have implemented it in Pyro with code available at `github.com/treigerm/sdvi_neurips`. The first baseline we consider, **Pyro AutoGuide**, uses the `AutoNormalMessenger` class to automatically generate a guide, and trains it with Pyro's built-in tools for VI (`http://pyro.ai/examples/svi_part_i.html`). As an additional VI baseline, we also implement a custom guide for each model which uses the variable-by-variable scheme outlined in Sec. 3, in combination with the score function gradient estimator; we refer to this baseline as **BBVI**. For SDVI, we run SH until there are 10 active SLPs left (i.e. $m = 10$ in Algo. 1) and parallelize the computation across 10 cores. We further construct each local guide distribution $q_k$ as a mean-field normal. The specific configurations for each method for each experiment are provided in App. D.

## 6.1   Program with Normal Distributions

We use our first experiment to further clarify the failure modes of existing VI approaches. We consider an extension of the model from Fig. 1 to contain more SLPs. The full model is

$$
\begin{aligned}
u &\sim \mathcal{N}(0, 5^2), \\
x &\sim \mathcal{N}(z, 1), \qquad \text{where} \qquad z = \begin{cases} 0, & \text{if } u \in (-\infty, -4] \\ K, & \text{if } u \in (-5 + K, -4 + K] \text{ for } K = 1, \dots, 8 \\ 9, & \text{if } u \in (4, \infty) \end{cases} \\
y &\sim \mathcal{N}(x, 1).
\end{aligned}
\tag{15}
$$

We assume we have observed $y = 2$. The results in Fig. 2 demonstrate that SDVI is able to overcome the limitations of the other variational approaches. BBVI and Pyro AutoGuide both use the same guide in this model; BBVI uses the score function gradient estimator for training, whereas Pyro AutoGuide uses the reparameterized gradient estimator. This difference results in different posterior approximations for the different baselines. The BBVI guide tends to place all its mass on a single SLP and then provides a suitable approximation for only that one SLP, ignoring all the others. This explains the large standard deviations for the ELBO values in Fig. 2b as the ELBOs in different SLPs will converge to drastically different values. For Pyro AutoGuide the biased gradient estimates will train the variational approximation for variable $u$ to be close to the prior $\mathcal{N}(0, 5^2)$. SDVI is able to avoid the shortcomings of the baselines as it provides an overall better posterior approximation leading to larger ELBO values, i.e. lower KL divergences to the true posterior, and a more accurate weighting of the different SLPs (Fig. 2a).

Table 1: Log posterior predictive densitiy (LPPD), ELBO, and maximum a posteriori (MAP) estimate for $K$ for GMM model. Mean and standard deviation for LPPD and ELBO computed over 5 runs.

| Method | LPPD ($\uparrow, \times 10^3$) | ELBO ($\uparrow, \times 10^3$) | MAP $K$ |
|--------|--------|--------|--------|
| DCC | $-9842.90 \pm 3904.57$ | N/A | 14, 11, 16, 14, 15 |
| BBVI | $-2217.07 \pm 146.31$ | $-8770.55 \pm 544.95$ | 25, 25, 25, 25, 25 |
| SDVI | $\mathbf{32.84 \pm 0.02}$ | $\mathbf{128.76 \pm 0.17}$ | 5, 5, 6, 6, 5 |
| S-SDVI | $\mathbf{32.80 \pm 0.02}$ | $\mathbf{128.63 \pm 0.22}$ | 5, 5, 6, 5, 6 |

## 6.2 Infinite Gaussian Mixture Model

Our next model is a Gaussian Mixture Model (GMM) with a Poisson prior on the number of clusters:

$$K \sim \text{Poisson}(9) + 1; \quad u_k \sim \mathcal{N}(\mathbf{0}, 10\,\mathrm{I}) \text{ for } k = 1, \ldots, K; \quad y \sim \frac{1}{K} \sum_{k=1}^{K} \mathcal{N}(\mu_k, 0.1\,\mathrm{I}),$$

where $I$ is the $D \times D$ identity matrix and $\mathbf{0}$ is a $D$ dimensional vector of zeros (we set $D = 100$). A similar model was considered in Zhou et al. [34] but with $D = 1$ instead of $D = 100$. We generate a dataset of 1250 observations with $K = 5$. To compare and evaluate the different algorithms, we hold out 250 data points as a test dataset to compute the log posterior predictive density (LPPD).

The Pyro AutoGuide baseline from the previous experiment is not applicable here since it assumes all latent variables are continuous. In BBVI, for practical reasons, we had to cap the maximum number of clusters in the guide at 25 (cf. App. D). To provide a further baseline, we have also implemented **DCC** [34] in Pyro with Random-walk lightweight Metropolis-Hastings (RMH) [63] as a local inference algorithm. We chose DCC in particular because it also exploits the same breakdown into SLPs, so comparing against DCC is an opportunity to highlight the benefits of using a VI method.

In this model, the observations are assumed to be conditionally independent given the latent variables, thus enabling SDVI to work on subsets of the whole dataset [19, 22]. Specifically, we run SDVI on a model which samples a random minibatch of size $B = 100$ at each iteration and then scales the likelihood by the factor $N/B$, where $N = 1000$ is the size of the full dataset; we refer to this setup as Stochastic SDVI (S-SDVI). Furthermore, for this model SDVI is able to directly construct valid local guides $q_k$ (using the mechanism for models branching on discrete variables outlined in Sec. 4.5) and therefore (S-)SDVI can use the reparameterized gradient estimator.

Table 1 shows that SDVI and S-SDVI significantly outperform the baselines, yielding a several orders of magnitude larger posterior predictive density and providing the only reasonable predictions for the numbers of clusters. In the few instances were (S-)SDVI returns a suboptimal MAP estimate of $K = 6$, this was because the local guide for the SLP with 5 components had fallen into a local model that fails to correctly identify all the clusters in the data, in turn returning a suboptimal local ELBO. BBVI and DCC struggle with this model due to the high-dimensional parameter space. DCC's local inference algorithm, RMH, only updates one variable at a time which results in slow mixing times. Note, DCC does not provide any ELBO values; its marginal likelihood estimator PI-MAIS [66] constructs an importance sampling (IS) proposal distribution based on the outputs of MCMC chains which could theoretically be used to estimate an ELBO value. However, as IS requires over-dispersed proposals, the ELBO scores for this approach are trivially $-\infty$, preventing a sensible comparison.

## 6.3 Inferring Gaussian Process Kernels

For our final experiment, we consider the problem of inferring the kernel structure of a Gaussian Process (GP). Following [67, 68], we place a prior over kernel functions using a probabilistic context-free grammar (PCFG) . We consider the squared exponential (SE), rational quadratic (RQ), periodic (PER), and linear (LIN) base kernels, and use the production rules

$$\mathcal{K} \to \text{SE} \mid \text{RQ} \mid \text{PER} \mid \text{LIN} \mid \mathcal{K} \times \mathcal{K} \mid \mathcal{K} + \mathcal{K}.$$

Sampling from the PCFG is implemented with a recursive probabilistic program that uses sam-

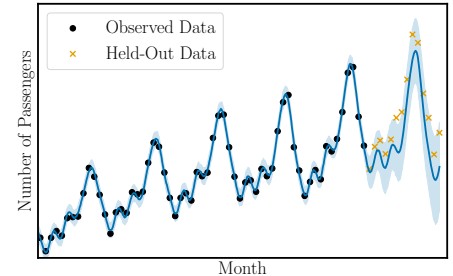

Figure 3: Posterior predictions of the GP for SDVI, shaded regions indicate 2 standard deviations that are computed from 100 posterior samples.

ples from a categorical distribution to decide which production rule in the PCFG should be applied. In addition to the kernel structure, we also perform inference over the kernel hyperparameters for each base kernel and the observation noise; we place an inverse-gamma prior on each base kernel hyperparameter and a half-normal prior on the observation noise. We further assume a normal likelihood function and marginalize out the latent GP. Additional model details are in App. D. We apply this model to a dataset of monthly counts of international airline passengers [69], withholding the last 10 % of all observations as a test dataset.

For SDVI we can construct local valid proposals $q_k$ using the mechanism for models with discrete branching outlined in Sec 4.5. Hence, in each SLP the local guide $q_k$ provides a posterior approximation over the kernel hyperparameters and the observation noise; the posterior distribution over kernel structures is implicitly defined through the mixture distribution over program paths. Table 2 shows that SDVI provides higher

Table 2: Final log posterior predictive densitiy (LPPD) and ELBO for GP model. Shown are mean and standard deviation computed over 5 runs.

| Method | LPPD ($\uparrow$) | ELBO ($\uparrow$) |
|--------|-------------------|-------------------|
| DCC  | $-58.92 \pm 32.47$ | N/A |
| BBVI | $-18.82 \pm 1.20$  | $-48.48 \pm 0.33$ |
| SDVI | $\mathbf{2.05 \pm 3.30}$ | $\mathbf{34.53 \pm 21.42}$ |

LPPD values, and is also able to achieve a higher final ELBO value compared to BBVI. Fig. 3 shows the posterior predictions for the SDVI run with the median LPPD score. SDVI is able to provide qualitatively reasonable predictions, as the predictions follow the periodic trend in the observed data.

# 7 Discussion

We believe that SDVI provides a number of significant contributions towards the goal of effective (automated) inference for probabilistic programs with stochastic support, nonetheless it still naturally has some limitations. Perhaps the most obvious is that it, if there is a very large number of SLPs that cannot be easily discounted from having significant posterior mass, it can be challenging to learn effective variational approximations for all of them, such that SDVI is likely to perform poorly if the number becomes too large. Here, customized conventional VI or reversible jump MCMC approaches might be preferable, as they can be set up to focus on the transitions between SLPs, rather than trying to carefully characterize individual SLPs.

Another limitation is that our current focus on automation means that there are still open questions about how best to construct more customized guides within the SDVI framework. Here the breakdown into individual SLPs and use of resource allocation strategies will still often be useful, but changes to our implementation would be required to allow more user control and customization. For example, the discovery of individual SLPs using the prior is a potential current failure mode, and it would be useful to support the use of more sophisticated program analysis techniques (e.g. [45]).

A more subtle limitation is that the local inferences of each SLP can sometimes still be quite challenging themselves. If the true posterior places a lot of mass near the boundaries of the SLP, there can still be a significant posterior discontinuity, meaning we might need advanced local variational families (e.g. normalizing flows) and/or gradient estimators. Such problems also occur in static support settings and are usually much more manageable than the original stochastic support problem, but further work is needed to fully automate dealing with them.

Finally, variational methods are often used not only for inference, but as a basis for model learning as well. In principle, SDVI could also be used in such settings, but as described in App. F, there are still some hurdles that need to be overcome to do this in practice.

# 8 Conclusion

We have presented SDVI and shown that it is able to overcome the limitations of existing VI approaches for programs with stochastic support by using a novel guide structure that breaks the program down into SLPs with fixed support, rather than matching the original stochastic control flow. The structure of the variational family separates the ELBO into multiple independent inference problems which naturally motivates a divide-and-conquer style training procedure with explicit resource allocation. Experimentally we found that these innovations meant that SDVI was able to provide significant performance improvements over the previous state-of-the-art approaches.

## Acknowledgments and Disclosure of Funding

We would like to thank Yuan Zhou for useful discussions in the early stages of this project. Tim Reichelt is supported by the UK EPSRC CDT in Autonomous Intelligent Machines and Systems with the grant EP/S024050/1. Luke Ong would like to acknowledge funding from EPSRC UK and National Research Foundation Singapore NRF-RSS2022-009.

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
