# Appendix for Rethinking Variational Inference for Probabilistic Programs with Stochastic Support

**Tim Reichelt**[1]     **Luke Ong**[1,2]     **Tom Rainforth**[1]

[1] University of Oxford
[2] Nanyang Technological University, Singapore
{tim.reichelt,lo}@cs.ox.ac.uk    rainforth@stats.ox.ac.uk

## A  KL Divergence Derivation

### A.1  Breaking Down the Global ELBO

The global ELBO is given by

$$\mathcal{L}(\phi, \lambda) = \mathbb{E}_{q(x;\phi,\lambda)} \left[ \log \frac{\gamma(x)}{q(x;\phi,\lambda)} \right], \tag{16}$$

$$= \int_{\mathcal{X}} q(x;\phi,\lambda) \log \frac{\gamma(x)}{q(x;\phi,\lambda)} dx, \tag{17}$$

using the fact that the subsets $\mathcal{X}_k$ provide a partition of $\mathcal{X}$ we can write the integral as

$$= \sum_{k=1}^{K} \int_{\mathcal{X}_k} q(x;\phi,\lambda) \log \frac{\gamma(x)}{q(x;\phi,\lambda)} dx, \tag{18}$$

using the factorization of $q(x;\phi,\lambda)$ and the fact that for $x \in \mathcal{X}_k$ the program density satisfies $\gamma(x) = \gamma_k(x)$ we get

$$= \sum_{k=1}^{K} \int_{\mathcal{X}_k} q_k(x;\phi_k)q(k;\lambda) \log \frac{\gamma_k(x)}{q_k(x;\phi_k)q(k;\lambda)} dx \tag{19}$$

then using the fact that $q(k;\lambda)$ does not depend on $x$ we have

$$= \sum_{k=1}^{K} q(k;\lambda) \int_{\mathcal{X}_k} q_k(x;\phi_k) \log \frac{\gamma_k(x)}{q_k(x;\phi_k)} dx - \log q(k;\lambda), \tag{20}$$

which we can write concisely as

$$= \mathbb{E}_{q(k;\lambda)} \left[ \mathcal{L}_k(\phi_k) - \log q(k;\lambda) \right], \tag{21}$$

where

$$\mathcal{L}_k(\phi_k) := \mathbb{E}_{q_k(x;\phi_k)} \left[ \log \frac{\gamma_k(x)}{q_k(x;\phi_k)} \right].$$

### A.2  Optimal Setting of $q(k;\lambda)$

**Proposition 1.** *Let $L = \{\mathcal{L}_1, \ldots, \mathcal{L}_K\}$ be the set of local ELBOs, defined as per* (7), *where $L$ is countable but potentially not finite. If $0 < \sum_{k=1}^{K} \exp(\mathcal{L}_k) < \infty$, then the optimal corresponding $q(k;\lambda)$ in terms of the global ELBO* (6) *is given by*

$$q(k;\lambda) = \exp(\mathcal{L}_k) \Big/ \sum_{\ell=1}^{K} \exp(\mathcal{L}_\ell). \tag{8}$$

*Proof.* By the assumption that $0 < \sum_{k=1}^{K} \exp(\mathcal{L}_k) < \infty$, we have that $\exp(\mathcal{L}_k)/\sum_{k=1}^{K} \exp(\mathcal{L}_k)$ forms a valid probability mass function over $k \in \{1, \ldots, K\}$. We can therefore rewrite (7) as

$$\mathcal{L}(\phi, \lambda) = \mathbb{E}_{q(k;\lambda)} \left[ \log \frac{\exp(\mathcal{L}_k)}{\sum_{k=1}^{K} \exp(\mathcal{L}_k)} - \log q(k; \lambda) \right] + \log \sum_{k=1}^{K} \exp(\mathcal{L}_k) \tag{22}$$

$$= -\mathrm{KL} \left( q(k; \lambda) \,\|\, \frac{\exp(\mathcal{L}_k)}{\sum_{k=1}^{K} \exp(\mathcal{L}_k)} \right) + \log \sum_{k=1}^{K} \exp(\mathcal{L}_k) \tag{23}$$

Now as second term in the above is constant in $q(k; \lambda)$ and a KL divergence is minimized when the two distributions are the same, we can immediately conclude the desired result that the optimal $q(k; \lambda)$ is

$$q(k; \lambda) = \frac{\exp(\mathcal{L}_k)}{\sum_{\ell=1}^{K} \exp(\mathcal{L}_\ell)}. \tag{24}$$

$\square$

Additionally, from (23) it follows that for the optimal setting of the mixture distribution $q(k; \lambda)$ the global ELBO is given by

$$\mathcal{L}(\phi, \lambda^*) = \log \sum_{k=1}^{K} \exp(\mathcal{L}_k).$$

## B  Details on Resource Allocation

### B.1  Background on Successive Halving

Successive Halving (SH) divides a total budget of $T$ iterations into $L = \lceil \log_2(K) \rceil + 1$ phases and starts by optimizing each of $K$ candidates, in our case the SLPs, for $\lfloor T/(KL) \rfloor$ iterations. It then ranks each of the candidates in terms of their performance, in our case the values of $\exp(\mathcal{L}_k)$, before eliminating the bottom half. This process then repeats, with each of the remaining candidates run for $2^{\ell-1}T/(KL)$ iterations at the $\ell$-th phase. This results in an exponential distribution of resources allocated to the different candidates, with more resources allocated to those that are more promising after intermediate evaluation.

Adapting it to our setting of treating the problem as a top-$m$ identification is done by simply using $L = \lceil \log_2(K) - \log_2(m) \rceil + 1$ phases instead of $L = \lceil \log_2(K) \rceil + 1$.

### B.2  Online Resource Allocation

Here, we present an online version of Algo. 1, where the term 'online' refers to the fact that the algorithm considers more and more SLPs as the computational budget increases. The online variant of the algorithm is useful if a user is unsure about the total iteration budget that they want to spend on the input program. This user might want to run SDVI with an initial iteration budget $T_1$ and after having observed the results, they might decide that they want to keep further optimizing the guide parameters. We therefore need to adapt Algo. 1 so that it can be 'restarted' after it has terminated. A naive approach to this would be to simply run Algo. 1 again but re-use the $q_k$'s for the SLPs that have already been discovered and only initialize the $q_k$ from scratch for SLPs which have not been seen before. However, this scheme is limited as it disproportionately favours SLPs which were discovered in the previous run. This is because for those SLPs the local ELBOs will already be relatively large compared to the newly added SLPs. As a consequence, SH will not assign significant computational budget to the SLPs that were added after the algorithm was restarted.

To safeguard against this behaviour we instead propose an online version of SDVI in Algo. 2 which is using a modified 'reward' for SH. Instead of ranking the different SLPs according to $\mathcal{L}_k(\phi_k(t_k))$ we instead propose the objective $\exp(\alpha \mathcal{L}_k(\phi_k(t_k)))/t_k$ where $0 < \alpha \leqslant 1$. The reward is scaled by the reciprocal of $t_k$ because we are no longer aiming to select the SLPs with the highest $\mathcal{L}_k(\phi_k(t_k))$ but instead aim to choose the SLPs which have been 'underselected' compared to other SLPs, assuming we should have selected them in proportion to $\exp(\alpha \mathcal{L}_k(\phi_k(t_k)))$. The scaling by the scalar $\alpha$ is a

further mechanism to encourage more exploration, with setting $\alpha = 0$ equivalent to uniform sampling in the limit of repeated SH runs. Since this adapted objective takes into account the computational budget that was spent on each SLP, it is a more suitable objective when running SH repeatedly.

---

**Algorithm 2** Online SDVI

---

**Require:** Target program $\gamma$, iteration budget per SH run $T$, minimum no. of SH candidates $m$, parameter controlling $\alpha > 0$ exploration
1: Extract SLPs $\{\gamma_k\}_{k=1}^K$ from $\gamma$ and set $\mathcal{C} = \{1, \ldots, K\}$
2: Formulate guide $q_k$ for each SLP and initialize parameters $\phi_k$
3: $t_k = 0$ for all $k \in \mathcal{C}$
4: **while** Stopping criteria not satisfied **do**
5: $\quad$ $\mathcal{C}' \leftarrow \mathcal{C}$
6: $\quad$ Phases in successive halving $L = \lceil \log_2(|\mathcal{C}|) - \log_2(m) \rceil + 1$
7: $\quad$ **for** $l = 1, \ldots, L$ **do**
8: $\quad\quad$ Number of iterations $n_l = \lfloor \frac{T}{L|\mathcal{C}'|} \rfloor$
9: $\quad\quad$ **for** $k \in \mathcal{C}'$ **do**
10: $\quad\quad\quad$ Perform $n_l$ optimization iterations of $\phi_k$ targeting $\mathcal{L}_{\mathrm{surr},k}(\phi_k)$
11: $\quad\quad\quad$ Estimate $\mathcal{L}_{\mathrm{surr},k}(\phi_k)$ using Monte Carlo estimate of Eq. (11)
12: $\quad\quad\quad$ $t_k = t_k + n_l$
13: $\quad\quad$ **end for**
14: $\quad\quad$ Remove $\min(\lfloor |\mathcal{C}'|/2 \rfloor, |\mathcal{C}'| - m)$ SLPs from $\mathcal{C}'$ with the lowest $\exp(\alpha \, \mathcal{L}_{\mathrm{surr},k}(\phi_k))/t_k$
15: $\quad$ **end for**
16: $\quad$ Extract new SLPs from $\gamma$ and add them to $\mathcal{C}$, set $t_{k'} = 0$ for each new SLP with index $k'$
17: **end while**
18: Truncate $q_k$ outside of SLP support, $\mathcal{X}_k$, using Eq. (13)
19: Estimate each $\mathcal{L}_k(\phi_k)$ using Monte Carlo estimate of Eq. (7)
20: Calculate $q(k; \lambda)$ according to Eq. (8) and return $q(x; \phi, \lambda)$ as per Eq. (4)

---

# C  Details for Training Local Guides

## C.1  Density Estimation of the Prior

Before we can define the KL divergence we first have to carefully define global and local prior distributions We first define what we informally call the global 'prior' distribution of the program as the product of all the terms added to the program density by the `sample` statements

$$\pi_{\mathrm{prior}}(x_{1:n_x}) := \prod_{i=1}^{n_x} f_{a_i}(x_i | \eta_i). \tag{25}$$

However, here we are using the term prior only informally, since (25) is not a prior in the conventional Bayesian sense since the $\eta_i$ can be functions of the observed data $y$. Note that here $n_x$ in (25) is again a random variable since the raw random draws $x_{1:n_x}$ of the program do not necessarily have fixed length. Then similarly we define local 'prior' distributions

$$\pi_{\mathrm{prior},k}(x_{1:n_k}) := \frac{\mathbb{I}[x_{1:n_k} \in \mathcal{X}_k] \prod_{i=1}^{n_k} f_{A_k[i]}(x_i | \eta_i)}{Z_{\mathrm{prior},k}} = \frac{\mathbb{I}[x_{1:n_k} \in \mathcal{X}_k] \pi_{\mathrm{prior}}(x_{1:n_k})}{Z_{\mathrm{prior},k}}, , \tag{26}$$

where

$$Z_{\mathrm{prior},k} := \int_{\mathcal{X}} \mathbb{I}[x \in \mathcal{X}_k] \, \pi_{\mathrm{prior}}(x) dx. \tag{27}$$

Note that for our purposes we will never actually have to estimate $Z_{prior,k}$, we only defined it to ensure that $\pi_{prior,k}$ is a normalized density. This allows us to define the forward KL divergence which we would like to optimize with respect to $\phi_k$

$$\mathrm{KL}(\pi_{\mathrm{prior},k}(x) \parallel \tilde{q}_k(x; \phi_k)) = \mathbb{E}_{\pi_{\mathrm{prior},k}(x)} \left[ \log \frac{\pi_{\mathrm{prior},k}(x)}{\tilde{q}_k(x; \phi_k)} \right] \tag{28}$$

which we can rewrite as

$$= \mathbb{E}_{\pi_{\mathrm{prior},k}(x)} \left[ \log \pi_{\mathrm{prior},k}(x) \right] - \mathbb{E}_{\pi_{\mathrm{prior},k}(x)} \left[ \log \tilde{q}_k(x; \phi_k) \right]. \quad (29)$$

The first term is a constant with respect to $\phi_k$ and therefore does not affect the optimization

$$\propto \mathbb{E}_{\pi_{\mathrm{prior},k}(x)} \left[ -\log \tilde{q}_k(x; \phi_k) \right], \quad (30)$$

then by the definition of $\pi_{\mathrm{prior},k}(x)$ in Eq. (26) this is equivalent to

$$= -\frac{1}{Z_{\mathrm{prior},k}} \mathbb{E}_{\pi_{\mathrm{prior}}(x)} \left[ \mathbb{I}[x \in \mathcal{X}_k] \log \tilde{q}_k(x; \phi_k) \right]. \quad (31)$$

Finally, $Z_{prior,k}$ is a constant with respect to $\phi_k$ and can be dropped

$$\propto \mathbb{E}_{\pi_{\mathrm{prior}}(x)} \left[ -\mathbb{I}[x \in \mathcal{X}_k] \log \tilde{q}_k(x; \phi_k) \right]. \quad (32)$$

We can estimate the gradients of the objective in Eq. (32) using a Monte Carlo estimator

$$\nabla_{\phi_k} \mathbb{E}_{\pi_{\mathrm{prior}}(x)} \left[ -\mathbb{I}[x \in \mathcal{X}_k] \log \tilde{q}_k(x; k, \phi_k) \right] \approx \frac{1}{N} \sum_{j=1}^{N} \mathbb{I}[x^{(j)} \in \mathcal{X}_k] \nabla_{\phi_k} \log \tilde{q}_k(x^{(j)}; k, \phi_k) \quad (33)$$

where $x^{(j)}$ are raw random draws generated by executing the input program forward. These gradient estimates can then be used in a stochastic gradient descent optimization procedure. In our experiments, we generate a fixed set of $N$ samples and re-use the same set of samples for the entire optimization process. Other approaches are also possible such as periodically collecting a new set of samples and using local MCMC moves to collect samples instead of repeatedly sampling from the prior.

## C.2  Exploiting Program Structure: Discrete Branching Optimization

In practice, many user-defined programs have structural properties which can be exploited to construct a valid local guide directly and deterministically (without resorting to the stochastic mechanism described in Sec. 4.5). Specifically, consider the class of programs whose program paths are determined by variables sampled from discrete distributions. For these programs, we can assume that for each SLP ($k$th, say) there is an (ordered) set of indices $I_{\mathrm{branch}} \subset \{1, \ldots, n_k\} = I$ and a set of constants $r_{k,1}, \ldots, r_{k,|I_{\mathrm{branch}}|} \in \mathbb{Z}$ such that the local unnormalized densities are expressible as

$$\gamma_k(x_{1:n_k}) = \gamma(x_{1:n_k}) \prod_{l=1}^{|I_{\mathrm{branch}}|} \mathbb{I}\left[ x_{I_{\mathrm{branch}}[l]} = r_{k,l} \right]$$

where $I_{\mathrm{branch}}[j]$ means the $j$th element in $I_{\mathrm{branch}}$. It follows that we can construct densities for the $k$th SLP on a subset of variables in $x_{1:n_k}$ by eliminating all the variables given by indices $I_{\mathrm{branch}}$ (by instantiating them to constants). This is effectively equivalent to replacing the `sample` statements corresponding to the variables which influence the control flow with `observe` statements which induces a new program density that has the form

$$\tilde{\gamma}_k(x_{1:n'_k}) = \prod_{i=1}^{n'_k} f_{A_k[I'[i]]}(x_i | \eta_i) \prod_{l=1}^{|I_{\mathrm{branch}}|} f_{A_k[I_{\mathrm{branch}}[l]]}(r_{k,l} | \eta_l) \prod_{j=1}^{n_y} g_{b_j}(y_j | \phi_j) \quad (34)$$

where $I' := [1, \ldots, n_k] \setminus I_{\mathrm{branch}}$, and $n'_k := |I'|$. Furthermore, if all the remaining r.v. are continuous distributions with support in $\mathbb{R}$ (i.e. $\mathrm{supp}(f_{A_k[i]}) = \mathbb{R}$ for $i \in I'$) then $\tilde{\gamma}_k(x_{1:n'_k})$ itself has support in $\mathbb{R}^{n'_k}$. It is then straightforward to construct a guide $q_k$ with support in $\mathbb{R}^{n'_k}$ using existing methods, and we can get gradient estimates using the reparameterization gradient estimator (assuming there are no more discontinuities in $\tilde{\gamma}_k$).

To realize the discrete branching optimization in our Pyro implementation we allow users to annotate the `sample` statements which influence the branching. While it is in principle possible to automatically identify programs with discrete branching using program analysis, formalizing and implementing such a program analysis tool to work with arbitrary Pyro program would be a significant contribution in itself which is out of scope for this paper as we are focused on the statistical evaluation of SDVI. Specifically, the relevant `sample` statements within a Pyro program can be annotated as follows: `pyro.sample("x", dist.Poisson(7), infer={"branching": True})`. Our implementation of SDVI is then able to use these annotations to create the density $\tilde{\gamma}_k$ in (34).

# D  Additional Details for Experiments

For all experiments that rely on optimization we use the Adam optimizer [70]. The experiments were executed on an internal cluster which uses a range of different computer architectures.

## D.1  Model From Figure 1

Listing 1: Pyro Code for Figure 1.

```python
import pyro
import pyro.distributions as dist

def model():
    x = pyro.sample("x", dist.Normal(0, 1))
    if x < 0:
        z1 = pyro.sample("z1", dist.Normal(-3, 1))
    else:
        z1 = pyro.sample("z2", dist.Normal(3, 1))

    x = pyro.sample("x", dist.Normal(z1, 2), obs=torch.tensor(2.0))

guide = pyro.infer.autoguide.AutoNormalMessenger(model)
optim = pyro.optim.Adam({"lr": 0.01})
svi = pyro.infer.SVI(
    model, guide, optim, loss=pyro.infer.Trace_ELBO()
)

for j in range(2000):
    svi.step()
```

The full Pyro code for the model in Fig. 1, including automatically generating and training the guide is given in Listing 1. The code for BBVI and SDVI is provided in the code supplementary. For Pyro's AutoGuide and BBVI we run the optimization for 2000 iterations with a learning rate of 0.01. Similarly, for SDVI we have a total iteration budget of $T = 2000$ and use a learning rate of 0.01; we set the minimum number of SH candidates to $m = 2$

## D.2  Program with Normal Distributions

For SDVI, we use $10^3$ samples from the prior to discover SLPs. To train the local guides to place support within the SLP boundaries we collect $10^2$ samples per SLP and optimize the objective in Equation (12) for $10^3$ iterations. We run Algorithm 1 with a total budget of $T = 10^5$ with 5 particles for the ELBO and to estimate the final SLP weights we use $10^3$ samples per SLP. We use a learning rate of 0.01.

For Pyro AutoGuide, we run the optimization for $10^5$ steps with 1 ELBO particle. For BBVI, we run the optimization for $10^4$ steps with 10 ELBO particles. For both we use a learning rate of 0.01.

## D.3  Infinite Gaussian Mixture Model

For SDVI, we use $10^3$ samples from the prior to discover SLPs, run Algorithm 1 with a total budget of $T = 2 * 10^4$ with 10 particles for the ELBO and to estimate the final SLP weights we use $10^2$ samples per SLP. We use a learning rate of 0.1.

For BBVI, we run for $2 * 10^4$ iterations using 10 particles for the ELBO and a learning rate of 0.1. In the guide, we use a categorical distribution for number of components $K$ over the range $K \in [1, 25]$. We ran initial experiments with instead using a Poisson distribution paramterized by the rate but we found this leads to an explosion in the number of components in the guide which resulted in the program running out of memory. For each $\mu_k$ the variational approximation is a diagonal Normal distribution parameterized by the mean and the diagonal entries in the covariance matrix.

For DCC, we run for 200 iterations, at each iteration we run 10 independent RMH chains generating 10 samples and to get a marginal likelihood estimate we use PI-MAIS [66] which places a proposal

distribution (in our case a Gaussian) on the outputs of the RMH chains and samples from this proposal $M$ times; we set $M = 10$.

### D.4 Inferring Gaussian Process Kernels

#### D.4.1 Model Details

Our probabilistic context-free grammar for the kernel structure has the production rules

$$\mathcal{K} \to \text{ SE} \mid \text{RQ} \mid \text{PER} \mid \text{LIN} \mid \mathcal{K} \times \mathcal{K} \mid \mathcal{K} + \mathcal{K}. \tag{35}$$

with the production probabilities $[0.2, 0.2, 0.2, 0.2, 0.1, 0.1]$. On each base kernel hyperparameter we place an InverseGamma$(\alpha = 2, \beta = 1)$ prior. For each base kernel the specific hyperparameters we wish to do inference over are:[2]

- Squared Exponential (SE): Lengthscale
- Rational Quadratic (RQ): Lengthscale, Scale Mixture
- Periodic (PER): Lengthscale, Period
- Linear (LIN): Bias

Assuming we have $N$ observations with inputs $\mathbf{x} \in \mathbb{R}^N$ and outputs $\mathbf{y} \in \mathbb{R}^N$ our model can then be written as

$$\mathcal{K} \sim \text{PCFG}(), \quad \sigma \sim \text{HalfNormal}(0, 1), \quad \mathbf{y} \sim \mathcal{N}(0, \mathcal{K}(\mathbf{x}) + \sigma^2 \mathbf{I}) \tag{36}$$

where PCFG() samples a kernel (and its hyperparameters) from the probabilistic context-free grammar and $\mathcal{K}(\mathbf{x})$ is the $N \times N$ covariance matrix computed from kernel $\mathcal{K}$.

#### D.4.2 Algorithm Configurations

For SDVI, we use $10^3$ samples from the prior to discover SLPs, run Algorithm 1 with a total budget of $T = 10^6$ with 1 particles for the ELBO and to estimate the final SLP weights we use $10^2$ samples per SLP. We use a learning rate of $0.005$.

For BBVI, we run for $10^5$ iterations using 10 particles for the ELBO and a learning rate of $0.005$. The guide uses a log-normal distribution for the kernel hyperparameters and the observation noise, and for the discrete variables which influence the kernel structure we use categorical distributions. For DCC, we run for $10^3$ iterations and otherwise use the exact same hyperparameters as in the Gaussian Mixture Model experiment.

## E Additional Experimental Results

### E.1 Program with Normal Distributions

For completeness we include here the results for DCC on the model from Sec. 6.1. DCC does not have the same fundamental limitations as the BBVI baselines therefore is competitive with SDVI and provides a similar squared error for the SLP weights. In fact, it is quite impressive that SDVI is able to match the performance of DCC because DCC leverages marginal likelihood estimators which asymptotically converge to the true marginal likelihood whereas SDVI calculates the weights based on the ELBO. This

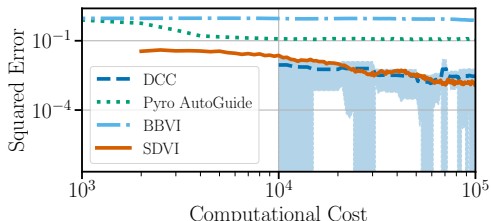

Figure 4: Squared error for the model in § 6.1 with DCC baseline. Conventions as in Fig. 2a.

is therefore a further indicator that for this model SDVI is able to provide good posterior approximations for each SLP.

---

[2]We use the same naming conventions as the Pyro Docs at `https://docs.pyro.ai/en/stable/contrib.gp.html#module-pyro.contrib.gp.kernels`.

# F  Difficulties of Parameter Learning for Models with Stochastic Support

In static support settings, one often uses variational bounds not only as a mechanism for inference, but also for training model parameters themselves [24, 41]. Using our notation from Sec. 2.2, this setting corresponds to having model parameters, $\theta$, that we wish to optimize alongside the variational parameters, $\phi$, such that the unnormalized density can be written as $\gamma(x; \theta)$, with corresponding normalization constant $Z(\theta)$. The ELBO then depends on both the variational and model parameters $\mathcal{L}(\phi, \theta) := \mathbb{E}_{q(x;\phi)}\left[\log \gamma(x;\theta)/q(x;\phi)\right]$. Provided $Z(\theta)$ is differentiable with respect to $\theta$, both $\phi$ and $\theta$ can then, at least in principle, be simultaneously optimized using stochastic gradient ascent.

However, similar as to the case of pure inference, naively extending this scheme to models with stochastic support is non-trivial and quickly runs into both conceptual and practical problems.

Parameters, $\theta_k$, that are inherently local to only a single SLP can be dealt with straightforwardly: as $\nabla_{\theta_k}\mathcal{L}_\ell = 0 \ \forall \ell \neq k$ for such parameters, we can simply ignore parameters not associated with the SLP we are updating, that is we only take a gradient step for $\{\phi_k, \theta_k\}$ on Line 5 of Algo. 1.

Problems start to occur, though, in the more common scenario where parameters are shared between SLPs, in the sense that they influence more than one $\gamma_k$. Consider, for example, the GP model from Sec. 6.3 and assume that instead of doing inference over the observation noise, $\sigma$, we instead wish to treat this as a learnable parameter instead. Here $\sigma$ could be seen as a 'global' model parameter as it appears in every SLP, so could be viewed as shared between them.

This now creates an issue in 'balancing' updates from different SLPs; the need to learn a shared $\theta$ breaks the separability between inference problems for individual SLPs. Consequently, we can no longer directly treat how often we update each SLPs as just a resource allocation problem: making more updates on a given SLP now increases the influence that SLP has on the $\theta$ which are learned. This problem is unlikely to be insurmountable—one could maintain a running estimate of $q(k; \lambda)$ during training and then use this to either directly control the resource allocation or scale the updates of $\theta$ depending on how often the corresponding SLP has been used—but it does represent a notable complication that would require its own careful consideration.

Beyond this specific practical challenge, there is also a more fundamental and general issue for parameter learning under stochastic support: should shared parameters be treated globally when we are learning them? Going back to the example of the observation noise, $\sigma$, in our GP example, it will actually be quite inappropriate here to learn a single global value for $\sigma$, as the optimal observation noise will be different depending on the kernel structure. Thus, though the variable is shared between SLPs in the program itself, it would be advantageous to learn separate values for it for each SLP, regardless of the inference approach we take.

The natural solution to this issue would be to perform parameter learning separately for each SLP, e.g. learning a separate $\sigma_k$ for each SLP in the GP example above. However, this raises a variety of issues in its own, not least the fact that the inference algorithm will now start to influence the model itself: SDVI and BBVI will learn fundamentally different models. There may also be settings where it is important for a parameter to be truly global and thus shared across the SLPs, e.g. because such sharing is an explicit prior assumption we wish to make.

Further problems occur when we consider that it is also feasible for learnable parameters to influence the control flow of the program, or even the set of possible SLPs. For example, a learnable parameter could impact the maximum possible recursion depth of a recursive program. This will create challenging interactions between SLPs: updates of one will influence the desirable behavior for the variational approximation of another. In turn, this can substantially complicate the resource allocation process and even the SLP discovery process itself.

Together, these aforementioned issues demonstrate that parameter learning for models with stochastic support is a complex issue, requiring specialist consideration beyond the scope of the current paper.

# G  Issues with Directly Training $q_k$

A natural question one might ask with the SDVI method is why do we not directly train $q_k$ to (7) by treating it as an implicit variational approximation defined by $\tilde{q}_k$? Namely, we can express (7) in

terms of $\tilde{q}_k$ as follows

$$\mathcal{L}_k(\phi_k) = \log \tilde{Z}_k(\phi_k) + \frac{1}{\tilde{Z}_k(\phi_k)} \mathbb{E}_{\tilde{q}_k(x;\phi_k)} \left[ \mathbb{I}\left[x \in \mathcal{X}_k\right] \log \frac{\gamma_k(x)}{\tilde{q}_k(x;\phi_k)} \right], \tag{37}$$

which, in principle, could be directly optimized with respect to $\phi_k$.

There are unfortunately two reasons that make this impractical. Firstly, though $\tilde{Z}_k(\phi_k)$ can easily be estimated using Monte Carlo, we actually cannot generate conventional unbiased estimates of $\log \tilde{Z}_k(\phi_k)$ and $1/\tilde{Z}_k(\phi_k)$ (or their gradients) because mapping the Monte Carlo estimator induces a bias. Second, this objective applies no pressure to learn a $\tilde{q}_k$ with a high acceptance rate, i.e. which actually concentrates on SLP $k$, such that it can easily learn a variational approximation that is very difficult to draw truncated samples from at test time.

By contrast, using our surrogate objective in (11) allows us to produce unbiased gradient estimates. Because of the mode seeking behaviour of variational inference, it also naturally forces us to learn a variational approximation with a high acceptance rate, provided we use a suitably low value of $c$. If desired, one can even take $c \to 0$ during training to learn an approximation which only produces samples from the target SLP without requiring any rejection. Figure 5 shows that empirically we learn a $\tilde{q}_k$ with a very high acceptance rates for the problem in Section 6.1.

Note that the surrogate and true ELBOs are exactly equal for any variational approximation that is confined to the SLP (as these have $\tilde{Z}_k(\phi_k) = 1$). This does not always necessarily mean that they have the same optima in $\phi_k$ for restricted variational families, even in the limit $c \to 0$. However, such differences originate from the fact that the trunctation can itself actually generalize the variational family (e.g. if $\tilde{q}_k$ is Gaussian, then $q_k$ will be a truncated Gaussians). As such, any hypothetical gains from targeting (7) directly will always be offset against drops in the acceptance rate of the rejection sampler.

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

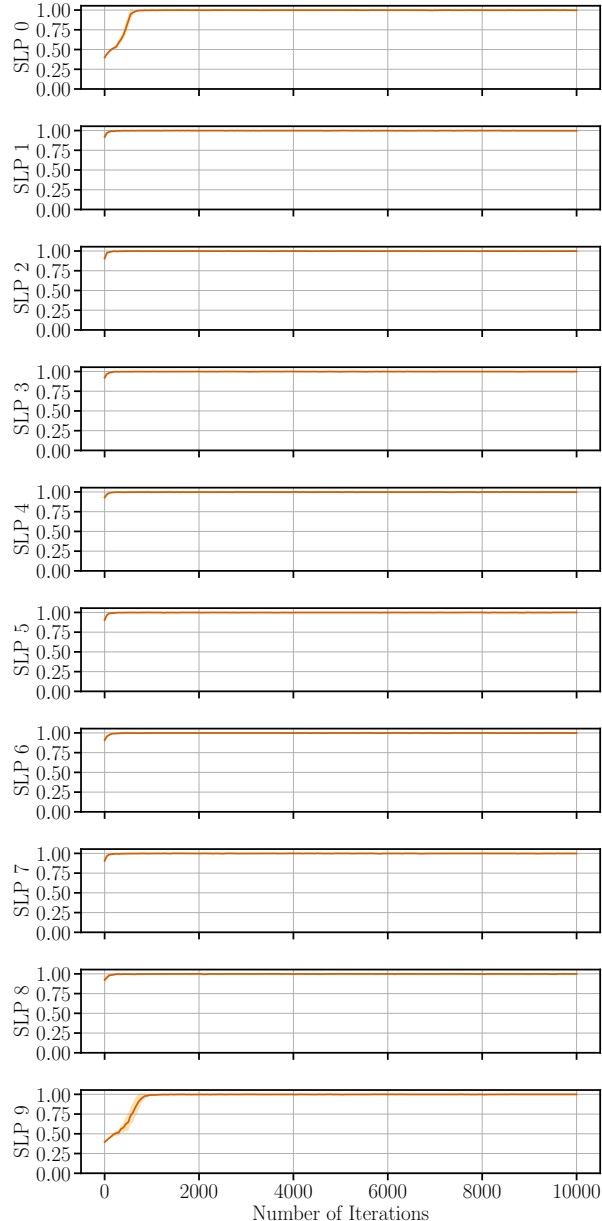

Figure 5: Acceptance rates for evaluating the local ELBOs in each SLP for the model from Sec. 6.1. Each plot represents a separate SLP; the plot with "SLP i" corresponds to the SLP with $z = i$ in Eq. (15). We can see that for all SLPs the acceptance rate approaches 1 with more iterations, confirming the mode seeking behaviour that arises when maximizing the surrogate ELBO in Eq. (11).