# OpenReview forum: "Rethinking Variational Inference for Probabilistic Programs with Stochastic Support"
_NeurIPS.cc/2022/Conference — NeurIPS 2022 Accept_

### Official Review · Reviewer_Efev · 2022-07-07

**Rating:** 8
**Confidence:** 5
**Soundness:** 4 excellent
**Presentation:** 4 excellent
**Contribution:** 3 good

**Summary:**

The authors address the problem of designing variational proposals for universal probabilistic programs with stochastic support.  Their method breaks down the stochastic support of the target program into a mixture model over control-flow paths through a space of straight-line programs, and the authors explain that their variational proposal method contrasts with Divide-Conquer-Combine (its closest cousin, used for MCMC sampling).  They demonstrate their method's performance on a toy program, an infinite mixture model, and learning Gaussian process kernels in a PCFG.  Their results appear to clearly set a new state of the art in log-predictive density and ELBO.

**Questions:**

When comparing against sampling-based inference methods, why not include one based on importance sampling?  Resample-move SMC inside DCC, for instance, could provide an estimate of the normalizing constant, and therefore (in log space) an ELBO to which to compare.

I don't understand what's meant by "marginal likelihood" for an SLP within a larger program.  Not every straight-line subprogram is going to make observations before reaching a site of stochastic control-flow, and so your importance weight could just be a prior/proposal density ratio.  Is that ratio's expectation used as a normalizing constant to allocate compute power?

The authors have now clarified and explained away my previous concerns.

**Limitations:**

The authors, peculiarly for a Pyro paper, make no actual use of neural networks to represent conditional densities.

**Strengths And Weaknesses:**

Strengths:
* Variational inference in deep universal PPLs is widely applicable and significant to a broad variety of downstream problems
* Clearly superior experimental results, with 100*K dimensionalities
* Latter two experiments are clearly nontrivial for variational inference settings
* Color-coding provides clarity
* Explanations and mathematics are clear to PPL readers

Weaknesses:
* Explanations and mathematics will be unclear to readers without probabilistic ML background

---

> ### Author Response · Authors · 2022-08-02
> **Response to Reviewer Efev**
>
> Thank you very much for your thoughtful review and positive feedback. We respond to your questions individually below.
>
> > Experiments are strong but limited to two low-dimensionality problems.
>
> We would like to point out that our IGMM experiment deals with far higher dimensionalities than have ever been previously considered for programs with stochastic support.  Specifically, the dimensionality of the parameter space $\{u_1,\dots,u_K\}$ for the SLP with $K$ components is $100 * K$, such that some of them have over a $1000$ dimensions.  By comparison, the analogous IGMM experiment in the DCC paper [35] only has dimensionality $1 * K$.
>
> > When comparing against sampling-based inference methods, why not include one based on importance sampling? Resample-move SMC inside DCC, for instance, could provide an estimate of the normalizing constant, and therefore (in log space) an ELBO to which to compare.
>
> The DCC baseline we are comparing to is actually already leveraging a marginal likelihood estimator based on importance sampling (IS), namely PI-MAIS [68] (referenced in the Appendix) which adaptively constructs an importance sampling proposal distribution based on the outputs of the MCMC chains. However, as IS requires overdispered proposals, the ELBO scores for this approach are trivially $-\infty$, preventing sensible comparison by this metric. We will use the extra space allowed for the camera-ready version to make this clearer in the main paper.
>
> Note also that we did not provide direct comparisons to more conventional IS baselines because the DCC paper had already shown substantial improvements over them. We could still easily add them if you think they are important though.
>
> > I don't understand what's meant by "marginal likelihood" for an SLP within a larger program. Not every SLP is going to make observations before reaching a site of stochastic control-flow, and so your importance weight could just be a prior/proposal density ratio. Is that ratio's expectation used as a normalizing constant to allocate compute power?
>
> Good question! While not every SLP will have observations, the boundaries of the SLP always place constraints on the density function. Specifically, $\gamma_k$ always contains the term $\mathbb{I}[x \in \mathcal{X}_k]$ which in itself is a form of conditioning as well. Therefore, inferring the boundaries of a given SLP can therefore be interpreted as its own inference problem as well, with the  $\gamma_k$ not being normalized densities even if we make no observations in that SLP.  The normalizing constant in such scenarios is simply the expectation of $\mathbb{I}[x \in \mathcal{X}_k]$ under the forward generative model, that is the probability of a prior sample being on that path. Therefore, it is still necessary to train a $q_k$ for these SLPs and allocate resources according to their normalization constant appropriately.  We will add a comment about this to the final version of the paper.

---

> > ### Comment · Reviewer_Efev · 2022-08-08
> > **Thanking the authors for their responses**
> >
> > Thank you for clarifying.  I will take your responses here into account in revising my review.

---

> > > ### Author Response · Authors · 2022-08-08
> > > **Response to Reviewer Efev**
> > >
> > > Thank you for considering our comments. We are glad to see that you have increased our score as a result.

---

### Official Review · Reviewer_M2CX · 2022-07-11

**Rating:** 7
**Confidence:** 3
**Soundness:** 3 good
**Presentation:** 3 good
**Contribution:** 3 good

**Summary:**

The paper proposes a variational inference method named Support Decomposition Variational Inference (SDVI) that targets probabilistic programs with stochastic support. The main idea is to perform variational inferences separately on sub-programs with static support. The variational approximation has a mixture form where each variational component is optimized independently for each sub-program. Finally, the approach is compared to baseline algorithms on a Gaussian model with stochastic control flow, on an infinite Gaussian mixture model (IGMM) with a stochastic number of clusters, and on inferring the kernel structure of a Gaussian Process.

**Questions:**

The “rethinking” word in the title seems to create confusion because the main algorithm is only compared to the simple Gaussian approximation.

The mention of Pyro’s AutoGuide in Figure 1 may cause confusion. It might be better to use an explicit algorithm name, e.g. Gaussian variational approximation or the like.

It is not clear to me how we can differentiate between different paths of a program. Based on the program in Figure 1, apparently it requires users to use different variable names (z1, z2) for different paths. Is this a requirement for users?

In the discussion of Table 1, “MAP estimate of K=6, the local guide q_k corresponding to the SLP with 5 components” - should this be SLP with 6 components?

**Limitations:**

Nothing to report.

**Strengths And Weaknesses:**

The paper is clearly written and the structure is easy to follow. I don’t have problems getting the main ideas.

The paper tackles the universality perspective in probabilistic programming systems. This seems to be a difficult problem. Though the main idea follows the “Divide, Conquer, and Combine” approach proposed by Zhou et al. [35], using variational inference instead of MCMC poses different challenges, which are reasonably addressed in the section 4:
+ Using the Successive Halving algorithm to avoid wasting computational resources on less interesting sub-programs.
+ Truncating the target density by a small positive value to ensure obtaining a finite ELBO (when a variational component proposes samples outside of the support of its corresponding sub-program).
+ Variational components need to be initialized smartly so that they have sufficiently large probability mass on the support of the corresponding sub-programs.

Despite that the authors already provide an implementation in Pyro, I feel that it is a bit tricky for practitioners to adopt the approach due to the complication of the above challenges.

The paper lacks discussions on how expensive the approach is (though it seems that the method is scalable through the IGMM experiment). This is important for users who want to apply the method in practice.

In addition, though the experimental results seem to be correct, it would be more interesting for readers to know in which situations we need to construct a probabilistic program with stochastic support. This might deserve a paragraph in the introduction section.

---

> ### Author Response · Authors · 2022-08-02
> **Response to Reviewer M2CX**
>
> Thank you very much for your careful review and helpful suggestions. We are glad to see that you are already in favour of acceptance and hope that our response addresses your remaining concerns.
>
> > The paper lacks discussions on how expensive the approach is (though it seems that the method is scalable through the IGMM experiment).
>
> Good point, we will add more direct discussion of this in the final version of the paper (utilising the fact we will then have an extra page available).  Here we would first like to point out that all our comparisons are cost-normalized—noting that the per-likelihood-evaluation cost of each approach is essentially identical—so the gains shown are in real-time terms. In fact, because SDVI allows parallelization of computation over the different SLPs in a manner that cannot be exploited by conventional VI approaches, our comparisons are thus actually very conservative, as they do not account for the speed ups this parallelization can provide.
>
> There is naturally a storage overhead for SDVI in that we learn multiple variational approximations instead of just one, but given the memory costs from the variational approximations are generally very low anyway, it will be very rare that this is a problem in practice.  SDVI also requires some extra computations, compared with standard VI, in evaluating the ELBOs themselves, required for resource allocation and calculating $q(k;\lambda)$. However, this can be very directly controlled and easily kept low (we use about 10% of computational budget for this in our experiments), while this cost is already accounted for by the cost normalization in our experiments.
>
>
> > Though the experimental results seem to be correct, it would be more interesting for readers to know in which situations we need to construct a probabilistic program with stochastic support. This might deserve a paragraph in the introduction section.
>
> Great suggestion, we will happily add this in for the final version of the paper. Many use cases arise from Bayesian Nonparametrics [Orbanz and Teh, "Bayesian Nonparametric Models," Encyclopedia of machine learning 1 (2010); Bloem-Reddy et al. “Sampling and inference for discrete random probability measures in probabilistic programs.” AABI (2017)]. Another common class of problems occur when performing model selection or Bayesian model averaging (or combination), as is often done in automatic data modelling (see e.g. [Saad et al. "Bayesian synthesis of probabilistic programs for automatic data modeling" POPL (2019)]). Furthermore, many scientific simulators, as used extensively in the physical sciences for example, define models with stochastic support because they often contain stochastic control flow. Since these simulator models are often very sophisticated and encode domain knowledge which has been accumulated over decades, scientists often want to use these simulator to make parameter inferences [15-17].
>
> > The “rethinking” word in the title seems to create confusion because the main algorithm is only compared to the simple Gaussian approximation.
>
> The “rethinking” in the title is meant to refer to the fact that we are constructing the guide in quite a fundamentally different way than is usually done for VI in PPLs (cf Section 3). This change is independent of the local form of approximations used for individual sample draws.
> Note that the guides we compare do not exclusively rely on Gaussian approximations (see Appendix D for details). We have made edits to try and make this clearer, but would be open to amending the title if you think it is still a significant source of confusion.
>
> > The mention of Pyro’s AutoGuide in Figure 1 may cause confusion. It might be better to use an explicit algorithm name, e.g. Gaussian variational approximation or the like.
>
> We agree this was potentially confusing. Another reviewer had an issue with this for a slightly different reason and so we have simply opted to remove this line from Figure 1.
>
> > Based on the program in Figure 1, apparently it requires users to use different variable names (z1, z2) for different paths. Is this a requirement for users?
>
> This is not a requirement on an algorithmic level, but our current implementation does indeed require that users give different addresses to each lexical sample site in the program if they want them to be treated as separate paths. This is inherited from a deliberate design decision in Pyro to let users assign addresses to each sample site manually, but many other PPLs (e.g. Anglican) will instead automatically assign unique variable identifiers for different paths. We will add discussion on this to the final paper.

---

> > ### Author Response · Authors · 2022-08-02
> > **Response to Reviewer M2CX (continued)**
> >
> > > ​​In the discussion of Table 1, “MAP estimate of K=6, the local guide q_k corresponding to the SLP with 5 components” - should this be SLP with 6 components?
> >
> > Thank you for highlighting this source of confusion. The “5 components” is actually correct, but we agree this sentence could have been clearer. What we are trying to say is that on the runs where SDVI returns an (incorrect) MAP estimate of $K=6$, this has occured because the local inference for the SLP with $5$ components has failed to train effectively. We have clarified this in the paper.

---

> > ### Comment · Reviewer_M2CX · 2022-08-08
> > **Response to authors**
> >
> > Thank you a lot for the clarification! Though I am still a bit concern about the practical aspect of the approach and the challenges around it, I'm happy to increase the score to reflect my feeling about the impact of the work.

---

> > > ### Author Response · Authors · 2022-08-08
> > > **Response to Reviewer M2CX**
> > >
> > > Thank you for taking the time to respond to our comments! We are delighted to hear that you are increasing your score.

---

### Official Review · Reviewer_A7EP · 2022-07-11

**Rating:** 7
**Confidence:** 3
**Soundness:** 3 good
**Presentation:** 3 good
**Contribution:** 3 good

**Summary:**

The paper proposes a new variant of variational inference for probabilistic programs with stochastic support. To address the challenges stochastic support poses, this paper proposes a novel way of constructing a variational guide by breaking down the problem into sub-programs. The paper demonstrates that this approach results in improvements in inference performance.

**Questions:**

- Could authors justify the main differences between their work and that of for instance [35]?
- How does it compare to MCMC VI?

**Limitations:**

- This work seems a little incremental thus some justification is required to distinguish between this work and others.
- Choosing a variational family is always a crucial task when doing VI. I am not sure how it is compared to the MCMC VI models.

**Strengths And Weaknesses:**


Strengths:
1) The paper addresses a very important and useful problem and the proposed method seems to be a nice and useful addition to probabilistic programming languages. I especially appreciate that the code is made available.
2) The paper is very well written and easy to understand
3) The paper contains comprehensive experiments to demonstrate the effectiveness of the method.

Weakness:
1) The work seems to be incremental over [33-35] especially [35]. The authors need to justify how their work differs from the aforementioned references.

2) The paper needs to include more related works and explain the differences and similarities between the proposed method and related works in more detail. There has been a lot going on in this field and it begs a more comprehensive literature review.

---

> ### Author Response · Authors · 2022-08-02
> **Response to Reviewer A7EP**
>
> Thank you very much for your thoughtful review and praise of our work. We are glad to see that you are already in favour of acceptance and hope that our response addresses your remaining concerns.
>
> > 1. The work seems to be incremental over [33-35] especially [35]. The authors need to justify how their work differs from the aforementioned references.
>
> We are very happy to add additional discussion on how our work differs from these papers (note they are now references [32-34] in the updated draft but we use the old numbers below), we will use the extra space available for the camera ready to expand our related work section to do this, based on the additional discussion below.
> Perhaps the biggest distinguishing technical factor to [35] is that our approach is a variational inference approach rather than a Monte Carlo approach, with a lot of technical challenges arising from this distinction. As Reviewer M2CX elegantly puts it:
> “[U]sing variational inference instead of MCMC poses different challenges, which are reasonably addressed in the section 4” (please also see their subsequent bullet points).  The practical significance of this change is borne out in our empirical results, where we see that the improved speed and scaling of variational inference leads to very substantial performance gains.  We believe our work also has novelty in showing how using a variational family based on SLPs naturally leads to divide-and-conquer style algorithms emerging, due to the resulting separability of the ELBO.
> [33] and [34] both also use the general idea of breaking down programs into SLPs, but both papers consider starkly different problem settings and have very different aims to our own work. [33] consider a restricted programming language which only allows linear arithmetic and only allows “hard” conditioning (i.e. condition on whether a certain condition evaluates to True/False). They therefore consider completely separate types of inference problems. Similarly, the language considered in [34] does not contain any “observe” statements at all and therefore does not consider the task of posterior inference. Neither have any direct link to variational inference.
>
>
> > How does it compare to MCMC VI?
>
> Existing work on MCMC VI has been focused on the setting of static support and we are not aware of any current approaches that can be applied in the stochastic support setting to provide a testable baseline we can compare to.
>
> The fact that SDVI breaks down the inference problems into sub-problems with static support means that it could provide a useful stepping stone to generalize MCMC VI techniques to the stochastic support setting, but investigating this interesting avenue properly is unfortunately beyond the scope of the current paper.
>
> There are also some interesting mathematical links between MCMC VI and our use of rejection sampling in the ELBO, we thank you for bringing these up and will add discussion on them to the paper for the camera ready.  For example, in cases where the learned $\tilde{q}_k$ has a very low acceptance rate under rejection sampling, one might be able to use MCMC techniques to more efficiently generate samples from the learned $q_k$ instead. We note though that this is tangential to the common practice of using MCMC to enrich variational families in order to enhance training of the model parameters, which does not assist the learning of the variational approximation itself.
>
> > 2. The paper needs to include more related works and explain the differences and similarities between the proposed method and related works in more detail. There has been a lot going on in this field and it begs a more comprehensive literature review.
>
> We hope that our answers to your questions above were able to alleviate your concerns around the related work. We plan to use the extra page allowed for the camera-ready submission to provide a more extensive literature review, and we are happy to add further discussions of any particular work that you believe needs to be mentioned as well.

---

> > ### Comment · Reviewer_A7EP · 2022-08-08
> > **Response to Authors**
> >
> > Thank you for the clarifications. I have carefully read your response to all the reviews and I believe the authors have answered most of the concerns. I am leaning towards the acceptance of this paper and ill change my score to accept.

---

> > > ### Author Response · Authors · 2022-08-08
> > > **Response to Reviewer A7EP**
> > >
> > > Thank you very much for your response and we are glad that you found our clarifications helpful!

---

### Official Review · Reviewer_NHpY · 2022-07-11

**Rating:** 7
**Confidence:** 3
**Soundness:** 3 good
**Presentation:** 2 fair
**Contribution:** 3 good

**Summary:**

This paper presents a new way to automatically construct and train variational families for models expressed as general probabilistic programs.

Given a program $p$, it first generates many (prior) samples from $p$ to identify a collection of control flow paths that collectively have high mass under the prior. Then, a separate variational family is trained to target the restriction of the model to each control flow path. During training, control-flow paths with low ELBOs are periodically pruned, so that computation is focused on paths that appear to have relatively high posterior probability. Finally, the learned variational families are combined into a mixture, with mixture weights based on estimated ELBOs for each path.

Experiments show competitive performance against other fully automated inference strategies for probabilistic programs, on three inference problems.

**Questions:**

1. In Section 4.5, you describe two methods for resolving the issue that local guides may not 'stay within' the boundaries of their SLPs. The way I read it, you are applying both solutions (mixing the target with a constant density, and rejection sampling the guide)---but isn't only one necessary? If you are rejection sampling, why introduce unnecessary discontinuities into the target?

2. I am confused about how the rejection sampling in Section 4.5 is supposed to work. As you write, the density of the variational family now contains a "Z" term, equal to the probability that q produces a sample that is accepted. This Z term must be computed when estimating the ELBO and its gradient. (Though the notation does not make this clear, Z depends on the parameters phi, so Z cannot simply be dropped when training.) How do you compute Z and its gradient, which in general will be an intractable integral over the SLP's region (which could be strangely shaped)?

(Note: my score is only "borderline accept" because I am not sure I've understood the method / if I have I am not sure it's sound, and also because I have concerns about certain misleading aspects of the current presentation, described in "Strengths and Weaknesses." If these are addressed during the author response, I hope to revise my score to a more confident recommendation.)

**EDIT AFTER DISCUSSION:** The author response has addressed my concerns and I am increasing my score to a 7.

**Limitations:**

As stated in "Weaknesses," I think the limitations are not adequately discussed. The class of probabilistic programs in universal PPLs is huge: when should we expect this algorithm to work, and when should we expect it to fail? i.e., when might users still resort to custom guides in Pyro? What assumptions does the new method make (e.g., non-negligible posterior mass on an SLP implies non-negligible prior mass)?

**Strengths And Weaknesses:**

Strengths:

1. Convincing empirical results, showing that for a similar computational budget, the new method outperforms a few existing fully automatic inference methods.
2. The paper grows the body of evidence that extracting SLPs and tackling inference separately in each (as done e.g. in DCC) is a useful technique for fully automated inference in universal PPLs.
3. In models with discrete latent variables that influence control flow, gradient estimators are often based on REINFORCE, and a variant of the "explore/exploit" problem from reinforcement learning arises: without any gradient signal helping it to explore other branches of the control flow graph, the variational family may settle into 'exploiting' one suboptimal branch. This work presents a nice way of introducing some kind of 'exploration', by allocating some training time up front to many different branches, and pruning only if they do not look promising. This is a neat way of getting around the high variance of the REINFORCE estimator. (It's a bit similar to the strategy of exactly marginalizing the discrete variable, except maybe a bit less expensive in the long run, because if one path is not promising, it is pruned early.)
4. An implementation in Pyro is provided, which increases the likelihood that this work will have real impact for practitioners.

Weaknesses:

1. In Section 4.5, rejection sampling the guide changes the variational family's density, as the authors write, introducing a "Z" term that measures the probability that the guide generates a sample that lies within its assigned SLP. But this Z term, which depends on the guide's parameters and so must be considered during optimization, appears to be intractable, and the authors do not explain how it is computed when estimating the ELBO and the ELBO's gradient. If it is ignored, I am worried that the gradient estimates would be biased. Furthermore, it doesn't seem this feature is 'stress-tested' by the experiments, the latter two of which use guides that ensure no rejection is necessary.

2. I did not see any clear discussion of the method's limitations. At times, I thought the authors made misleading claims that downplayed limitations. For example, L176 says of the SLP discovery process that "experiments show it can reliably identify all SLPs with non-negligible posterior mass." But my understanding is that you only discover SLPs with non-negligible _prior_ probability mass; in all the experiments, the posterior was concentrated in SLPs likely under the prior. Consider fitting the Gaussian mixture model to a dataset with many more components than the prior expects. The posterior will concentrate far from the prior, in an SLP that may not even have been discovered by the algorithm. Of course, this is a difficult case for many inference algorithms, but the limitation should be clearly explained.

3. I find Figure 1 and its explanation on L99-110 misleading. The explanation makes it sound as though the problem in the example is the expressiveness of the default variational family (it cannot represent the posterior on x, due to stochastic control flow). But the striking behavior in the figure, where Autoguide fails, arises just because you are using the reparameterization trick in a non-reparameterizable model. This seems to be a totally separate question from the expressiveness of variational families: it's about whether the Pyro software happens to detect when to switch to the score function gradient estimator automatically. Unless your method *does* automatically detect when the reparameterization trick can or cannot be soundly applied, without any user intervention, this feels irrelevant to the paper. It reads as though a (predictable) failure of the gradient estimator is being passed off as a limitation of the variational family. This also applies to experiments that report results using biased gradient estimators—to me that feels like an orthogonal issue.

4. I don't think the introduction (e.g. L24-42) gives an accurate summary of the state of the art. It is true that many PPLs, including those listed here, aim to support variational inference, but the aim has *not* in general been to automate the construction of guides (although of course, that is an interesting goal). Although Pyro has an Autoguide feature, many of the other PPLs in the list do not: the *intended* use case is that a user constructs a model *and* a guide as probabilistic programs, and the system automates gradient estimation and optimization of the ELBO. (Thus, L92-98 are also a bit misleading.) Even in Pyro, which does support automatic guide construction, many of the tutorials show how to construct custom guides, and the flexibility to do so is touted as a key feature. Arguably, this is what makes languages like Pyro useful for a much broader variety of problems than earlier languages that attempt to fully automate inference. I like that this paper tackles the hard problem of fully automated inference, and I think it makes progress in that direction, but the scope of the problem within the broader field of probabilistic programming should be clearly described: many PPLs already support accurate variational inference (with correct gradient estimates) in models with stochastic control flow if the user provides a sensible variational family. This paper is attempting to further automate the inference process by automatically constructing guides for a broader class of models than existing autoguide techniques can handle.


Small nitpick: on line 63, I don't think "higher-order" is relevant -- a first-order language with branching and recursion yields the same class of models.

---

> ### Author Response · Authors · 2022-08-02
> **Response to Reviewer NHpY**
>
> Thank you very much for your extremely thorough and helpful review; you raise a number of excellent points that will help us  improve the paper.  We hope that our response below and paper update alleviate the concerns that prevented you from making a more confident positive recommendation.
>
> > Q1. [...] you describe two methods for resolving the issue that local guides may not 'stay within' the boundaries of their SLPs. The way I read it, you are applying both solutions (mixing the target with a constant density, and rejection sampling the guide)---but isn't only one necessary?
>
> This is a very good question that we agree was not properly addressed in the original submission. The short answer is that  we need both because we cannot easily use rejection sampling when training the individual variational approximations, but need the truncation when producing samples at test time and evaluating the ELBOs for resource allocation.
>
> More precisely, truncating the guide via rejection sampling is needed to ensure we produce valid samples for the target SLP, once the individual variational approximations have been learned. Without this, our guides might generate invalid samples at test time that cannot even be evaluated.  It is also needed for evaluating the ELBO for $q_k$ (as per the right hand side of Eq. (7)) for resource allocation and constructing $q(k;\lambda)$, noting that the true ELBO of $\tilde{q}_k$ is $-\infty$ if it places any mass outside the SLP.
>
> However, such truncation/rejection sampling is generally problematic (see below) when training the individual $\phi_k$. Here mixing the target with a constant density provides a method to avoid these issues by instead training the $\phi_k$ to Eq. (11) (note this previously had a typo which may have caused confusion: it should have been an expectation with respect to $\tilde{q}_k$, not $q_k$).
>
> The problems with rejection sampling during training relate to the fact that $\tilde{Z}_k$ depends on $\phi_k$, exactly as you allude to in Weakness 1 (W1), and the need to learn a rejection sampler with a high acceptance rate.  To see these issues, note that $\mathcal{L}_k(\phi_k)$ from Eq. (7) can be directly expressed in terms of $\tilde{q}_k$:
> $$\mathcal{L}_k(\phi_k) = \log \tilde{Z}_k(\phi_k) + \frac{1}{\tilde{Z}_k(\phi_k)} \mathbb{E}\_{\tilde{q}_k(x;\phi_k)}\left[ \mathbb{I}(x\in\mathcal{X}_k) \log \frac{\gamma_k(x)}{\tilde{q}_k(x;\phi_k)} \right].$$
> Now the dependency of $\tilde{Z}_k(\phi_k)$ on $\phi_k$ makes this an impractical objective for optimizing $\phi_k$. In particular, though $\tilde{Z}_k(\phi_k)$ can easily be estimated using Monte Carlo, we actually cannot generate conventional unbiased estimates of $\log \tilde{Z}_k(\phi_k)$ and $1/\tilde{Z}_k(\phi_k)$ (or their gradients) because mapping the Monte Carlo estimator induces a bias.  Furthermore, this objective applies no pressure to learn a $\tilde{q}_k$ with a high acceptance rate, i.e. which actually concentrates on SLP $k$, such that it can easily learn a variational approximation that is very difficult to draw truncated samples from at test time.
>
> By contrast, using our surrogate objective in Eq. (11) allows us to produce unbiased gradient estimates. Because of the mode seeking behaviour of variational inference, it also naturally forces us to learn a variational approximation with a high acceptance rate, provided we use a suitably low value of $c$ in Eq. (10).  If desired, one can even take $c \rightarrow 0$ during training to learn an approximation which only produces samples from the target SLP without requiring any rejection.  We have added a plot in Appendix G that shows that we do indeed learn very high acceptance rates for the problem in Section 6.1.
>
> Note that the surrogate and true ELBOs are exactly equal for any variational approximation that is confined to the SLP (as these have $\tilde{Z}_k(\phi_k)=1$).  This does not always necessarily mean that they have the same optima in $\phi_k$ for restricted variational families, even in the limit $c\rightarrow 0$.  However, such differences originate from the fact that the trunctation can itself actually generalize the variational family (e.g. if $\tilde{q}_k$ is Gaussian, then $q_k$ will be a truncated Gaussians).  As such, any hypothetical gains from targetting Eq. (7) directly will always be offset against drops in the acceptance rate of the rejection sampler.
>
> We appreciate that this was not properly explained in the original version of the submission and have added a new section in Appendix G to correct this. We have also made edits to the main paper to make it clearer that we do not use rejection sampling during training and the high level reasons for this.

---

> > ### Author Response · Authors · 2022-08-02
> > **Response to Reviewer NHpY (continued)**
> >
> > > Q2. [...] I am confused about how the rejection sampling in Section 4.5 is supposed to work. [...] How do you compute Z and its gradient?
> >
> > As mentioned above, the rejection sampling step is only used to evaluate the ELBO and sampling from the learned variational approximation, we don’t use it for training and therefore do not ever need to estimate its gradients.
> >
> > To be precise on how we estimate it, we first draw $N$ samples $\{x^{(i)}\}_{i=1}^N$ from $\tilde{q}_k$.  We then reject samples which do not fall into the SLP and estimate $\tilde{Z}_k$ as the acceptance rate of this sampler (i.e. $N_A/N$ where $N_A$ is the number of samples accepted).  Finally, using $A$ to denote the set of indices of accepted samples, we form our ELBO estimate as
> > $$
> > \hat{\mathcal{L}}_k = \frac{1}{N_A} \sum\_{i \in A} \log \frac{N_A \gamma_k (x^{(i)})}{N \tilde{q}_k(x^{(i)};\phi_k)}.
> > $$
> > Note here that $A$ and $N_A$ are random variables that both implicitly depend on $\phi_k$, which is why we can use this for estimation, but not training.
> >
> > We have added the explicit form of this estimator to the new section Appendix H.
> >
> > > W1. Rejection sampling [...] introduces a  Z term [...] which depends on the guide's parameters and so must be considered during optimization, appears to be intractable, and the authors do not explain how it is computed when estimating the ELBO and the ELBO's gradient.
> >
> > We believe this concern stems mostly from a misunderstanding: as previously alluded to, rejection sampling is not used in training the individual $\tilde{q}_k$ and so $\tilde{Z}_k$ (and its gradients) do not need to be calculated in the optimization of $\phi_k$. However, we appreciate that we could have been clearer in explaining how the local guides are constructed and trained, and have made adjustments to the paper to clarify this, including making the dependency of $\tilde{Z}_k$ on $\phi_k$ explicit.
> >
> > > W2. I did not see any clear discussion of the method's limitations. At times, I thought the authors made misleading claims that downplayed limitations.
> >
> > We are very sorry that you felt this, it was never our intention to be misleading about the limitations and we are very happy to make updates to ensure this is not the case.  We address some of your more specific points on this below, while we also plan to add a dedicated limitations section to the paper. Namely, utilising the fact that an extra content page is allowed for the camera ready (but not the revision now), we plan to add the following discussion:
> >
> > “While we believe that SDVI provides a number of significant contributions towards the goal of effective (automated) inference for probabilistic programs with stochastic support, it still naturally has some limitations. Perhaps the most obvious is that it if there is a very large number of SLPs that cannot be easily discounted from having significant posterior mass, it can be challenging to learn effective variational approximations for all of them, such that SDVI is likely to perform poorly if the number becomes too large. Here, customized conventional VI or reversible jump MCMC approaches might be preferable, as they can be set up to focus on the transitions between SLPs, rather than trying to carefully characterize individual SLPs.
> >
> > Another limitation is that our current focus on automation means that there are still open questions about how one would construct more customized guides within the SDVI framework. Here the breakdown into individual SLPs and use of resource allocation strategies will still often be useful, but changes to our implementation would be required to allow more user control and customization. For example, the discovery of individual SLPs using the prior is a potential current failure mode and it would be useful to support the use of more sophisticated program analysis techniques (like [45]).
> >
> > A more subtle limitation is that the local inferences of each SLP can sometimes still be quite challenging themselves. If the true posterior places a lot of mass near the boundaries of the SLP, there can still be a significant posterior discontinuity, meaning we might need advanced local variational families (e.g. normalizing flows) and/or gradient estimators. Such problems also occur in static support settings and are usually much more manageable than the original stochastic support problem, but futher work is needed to fully automate dealing with them.
> >
> > Finally, variational methods are often used not only for inference, but as a basis for model learning as well. In principle, SDVI could also be used in such settings, but as described in Appendix F, there are still a number of challenges that need to be overcome to do this in practice.”

---

> > > ### Author Response · Authors · 2022-08-02
> > > **Response to Reviewer NHpY (continued)**
> > >
> > > > W2 cont. [...] For example, L176 says of the SLP discovery process that "experiments show it can reliably identify all SLPs with non-negligible posterior mass." [...]
> > >
> > > We agree that this sentence should be rephrased and was a poor choice of words. We meant to convey that in the specific experiments we consider this method is able to identify SLPs that overall lead to non-negligible ELBO values, but you are right that this will not always be the case (such as in the example you suggest); we have made this clearer. We further note here that the mechanism to find SLPs is a modular part of our algorithm and  we will add some further discussion on when it might be necessary to switch to other methods like MCMC sampling [35] or static analysis of the program code [33,44,45].
> > >
> > >
> > > > W3. I find Figure 1 and its explanation on L99-110 misleading
> > >
> > > We apologize that these were not clear and agree they could be misleading; we were trying to make two separate points that have become inappropriately conflated when trying to summarize them in a single figure. The main goal of Fig. 1 and L99-110 is to highlight the shortfalls of BBVI (the blue line in Fig. 1). Here, BBVI is using the same guide form as the Pyro AutoGuide but with the bias in gradient estimates corrected. Note that BBVI also converges to a sub-optimal solution which is due to the form of the variational family.
> > >
> > > We included the Pyro AutoGuide method so that the readers can see the current default behaviour for programs with stochastic support. Noting that the high variance issues one often experiences using REINFORCE gradients can actually cause larger final errors than the bias of invalidly using reparameterization, the default behaviour of Pyro AutoGuide is not entirely gratuitous without a method like SDVI. As you say yourself, one of the advantages of our approach is in reducing variance and thus avoiding this problem trade-off, so we would argue the failures of Pyro AutoGuide are not completely tangential.  However, we appreciate that this link is very subtle and not the conclusion the reader will most naturally make. We have thus decided to just remove the Pyro AutoGuide from Fig 1 and have updated the writing accordingly.
> > >
> > > > W3 cont. This also applies to experiments that report results using biased gradient estimators—to me that feels like an orthogonal issue.
> > >
> > > We included the Pyro AutoGuide in the experiment in Section 6.1 primarily because it was actually outperforming BBVI despite the bias (presumably because this was outweighed by gains in variance) and so it felt inappropriate to omit it. We are happy to remove it if you still think it is a problem though.
> > >
> > > > W4. I don't think the introduction (e.g. L24-42) gives an accurate summary of the state of the art. It is true that many PPLs [...] aim to support variational inference, but the aim has not in general been to automate the construction of guides  [...] the scope of the problem within the broader field [...] should be clearly described: many PPLs already support accurate variational inference (with correct gradient estimates) in models with stochastic control flow if the user provides a sensible variational family. [...]
> > >
> > > We agree that there have been many significant advances in the design of PPL which allows users to express custom guides. We have tried to emphasise that our paper focuses on the case when the guide should be constructed automatically. We have made some initial edits to the introduction to make this clearer, and will use the extra space for the camera ready submission to provide a more extensive discussion of these systems.
> > >
> > > However, we would like to point out that it isn't standard practice to construct the custom variational families/guides based on decomposing the input program into SLPs: the process laid out at the start of Section 3 is still very much the norm for manually constructed guides. Moreover, even constructing an SLP-based variational family manually is actually very difficult without some of the techniques we have introduced here, such as our use of surrogate ELBOs and our resource allocation scheme. Therefore, our work is also relevant beyond the setting of automated inference because it provides evidence of the benefit of this decomposition and introduces many of the tools needed to utilize it.
> > >
> > > > Small nitpick: on line 63, I don't think "higher-order" is relevant -- a first-order language with branching and recursion yields the same class of models.
> > >
> > > We have updated the writing to reflect that. Thank you for pointing this out.

---

> > > > ### Comment · Reviewer_NHpY · 2022-08-09
> > > > **Thank you for the thorough response!**
> > > >
> > > > Thank you to the authors for the very thorough response. As I mentioned, my initial 'borderline' score was due primarily to my uncertainty that I had understood the algorithm, and particularly what role the rejection sampling played. Your explanation here is very clear. I have checked the modified text in the paper and I believe it could still be more clearly signposted:
> > > >
> > > > * On L240, maybe you could clearly signpost that you are introducing the surrogate target temporarily, solely for optimizing the local SLP.
> > > >
> > > > * More importantly, L262 is the bit I find confusing --- the "finally" makes it sound as though you are introducing a final modification to the SLP training procedure. I think it would help to explicitly explain at the start of this paragraph that the surrogate ELBO is only used during training, and that when estimating the local ELBO for pruning SLPs and setting the final mixture weights, a different solution is required.
> > > >
> > > > > We included the Pyro AutoGuide in the experiment in Section 6.1 primarily because it was actually outperforming BBVI despite the bias (presumably because this was outweighed by gains in variance) and so it felt inappropriate to omit it. We are happy to remove it if you still think it is a problem though.
> > > >
> > > > Thank you for the explanation. I do think it makes the most sense to compare to other unbiased gradient estimators---whether the bias of reparam will be acceptably low seems very dependent on the model, and outside the scope of the paper. But I understand your reasoning and am happy to leave this to your judgment.
> > > >
> > > > > Namely, utilising the fact that an extra content page is allowed for the camera ready (but not the revision now), we plan to add the following discussion: ...
> > > >
> > > > Thank you -- I think this discussion is nicely written.
> > > >
> > > > > However, we would like to point out that it isn't standard practice to construct the custom variational families/guides based on decomposing the input program into SLPs: the process laid out at the start of Section 3 is still very much the norm for manually constructed guides. Moreover, even constructing an SLP-based variational family manually is actually very difficult without some of the techniques we have introduced here, such as our use of surrogate ELBOs and our resource allocation scheme. Therefore, our work is also relevant beyond the setting of automated inference because it provides evidence of the benefit of this decomposition and introduces many of the tools needed to utilize it.
> > > >
> > > > Thanks, I agree, and I think the paper will benefit from acknowledging this explicitly.
> > > >
> > > > This very thorough author response has addressed my concerns and I will raise my score to a 7.

---

> > > > > ### Author Response · Authors · 2022-08-09
> > > > > **Author Response**
> > > > >
> > > > > Thank you very much for your positive response to our comments and increasing your score! We appreciate the time and effort that you have put into this review, we believe it has clearly helped to improve the paper.
> > > > >
> > > > > > Your explanation here is very clear. I have checked the modified text in the paper and I believe it could still be more clearly signposted [...]
> > > > >
> > > > > Thank you very much for your feedback. It is very valuable and will significantly improve the exposition of our method. We will incorporate your proposed changes and add extra discussion in L240 to highlight that the surrogate density is only used temporarily. Further, we will remove the “Finally” from L262 and emphasize that the surrogate ELBO is only suitable for optimizing the variational parameters.

---

> > > > ### Comment · Reviewer_NHpY · 2022-08-09
> > > > **One more comment**
> > > >
> > > > I think the revised introduction presents the state of affairs more accurately. One part that I still disagree with slightly is the claim that "existing approaches all use a single global guide that mirrors the control flow of the input program, then introduce a variational approximation for each unique variable." Read in context, "existing approaches" here refers to all guide construction methods (including manual guide construction), and I believe there are several prominent examples of more complex variational families that do not use mean-field approximations, and do not follow the same control flow as the model.
> > > >
> > > > For instance, consider the Attend Infer Repeat model (https://arxiv.org/pdf/1603.08575.pdf, Figure 2): in the model program, a geometrically distributed number is sampled, and that many objects are generated. However, in the guide program, a `while` loop is used, in which a recurrent network repeatedly proposes a new object until deciding to stop.
> > > >
> > > > Of course, AIR is quite different from your proposed method. However, I believe that in the probabilistic programming community, especially among those who work on languages with support for variational inference with custom guides, it is very much already a goal to support models like AIR, with variational families that are *not* mean-field and which handle control flow in a smart way.

---

> > > > > ### Author Response · Authors · 2022-08-09
> > > > > **Author Response**
> > > > >
> > > > > We agree that there are already more complex, custom variational families that were proposed for specific models with stochastic support and that the section you highlight could be more clear about this. We will emphasize more clearly that we are focusing on automated guide construction methods and we are going to add a discussion of the fact that bespoke guides with more complex structures have been developed for specific models and that existing PPSs such as Pyro provide users with tools to express these bespoke guides easily.

---

### Author Response · Authors · 2022-08-02
**General Response**

We would like to thank the reviewers for their thoughtful reviews which have all been clearly written with significant effort and care. We are delighted that all reviewers have backed acceptance and gave almost universally strong subscores on Soundness, Presentation, and Contribution. Further, we gladly notice that the reviewers appreciated our “convincing” (Reviewer NHpY) and “comprehensive” (Reviewer A7EP) empirical results that “clearly set a new state of the art” (Reviewer Efev); believe that “the paper addresses a very important” (Reviewer A7EP), “difficult” (Reviewer M2CX), and “significant” (Reviewer Efev) problem; provides a “neat” (Reviewer NHpY) method which  “seems to be a nice and useful addition to PPLs” (Reviewer A7EP) and which is “widely applicable and significant to a broad variety of downstream problems” (Reviewer Efev); and found the paper to be “very well written and easy to understand” (Reviewer A7EP).

We respond to the questions of each reviewer individually below. We would again like to thank the reviewers for their helpful suggestions, which we will happily incorporate and which we believe will further strengthen the paper. We have already uploaded an updated version of the paper, but please note that we will only be able to provide some of the planned updates in the final version of the paper, when we can utilise the fact that an additional page will be allowed (unlike now).

---

### Author Response · Authors · 2022-08-08
**Follow up**

Dear Reviewers and AC

Thank you again for all your hard work.

We just wanted to quickly follow up again about our rebuttal to the reviews as there is not long left for review-author discussions.  Please let us know if our responses have alleviated your concerns or if there is anything you would like further clarification or additional changes on.

Many thanks!

Paper 3173 Authors

---

### Meta-Review · Area_Chair_EATM · 2022-08-21

**Recommendation:** Accept
**Confidence:** Certain

**Metareview:**

The reviewers have reached consensus after processing the authors' feedback. They all agree that this manuscript presents an interesting approach to applying variational inference in a setting of probabilistic programming that is of interest to the community. The reviewers raise tangible points that the authors have incorporated into their revision. I recommend that the authors continue to polish their manuscript to clearly address these points in the final version of their manuscript.

**Award:**

No

---

### Decision · Program_Chairs · 2022-09-14

Accept